# Combination Ensemble and Explainable Deep Learning Framework for High-Accuracy Classification of Wild Edible Macrofungi

**DOI:** 10.3390/biology14121644

**Published:** 2025-11-22

**Authors:** Aras Fahrettin Korkmaz, Fatih Ekinci, Eda Kumru, Şehmus Altaş, Seyit Kaan Güneş, Ahmet Tunahan Yalçın, Mehmet Serdar Güzel, Ilgaz Akata

**Affiliations:** 1Faculty of Health Sciences Nutrition, Dietetics Department, Şirinevler Campus, İstanbul Kültür University, 34191 Istanbul, Türkiye; a.korkmaz@iku.edu.tr; 2Institute of Artificial Intelligence, Ankara University, 06100 Ankara, Türkiye; fatihekinci@ankara.edu.tr; 3Graduate School of Natural and Applied Sciences, Ankara University, 06830 Ankara, Türkiye; 4Department of Computer Engineering, Faculty of Engineering, Ankara University, 06830 Ankara, Türkiye; 5Artificial Intelligence and Data Engineering, Faculty of Engineering, Ankara University, 06830 Ankara, Türkiye; 6Department of Biology, Faculty of Science, Ankara University, 06100 Ankara, Türkiye

**Keywords:** edible mushroom, deep learning, ensemble models, explainable AI, species classification

## Abstract

This study evaluated 10 deep learning models on a dataset of 24 wild edible macrofungi species. Among single models, EfficientNetB0 reached 95.55% accuracy, while MobileNetV3-L performed lowest (90.55%). Pairwise ensembles showed limited gains, but the proposed Combination Model (EfficientNetB0 + ResNet50 + RegNetY) achieved the best results with 97.36% accuracy, 0.9996 AUC, and 0.9725 MCC. Explainable AI methods (Grad-CAM, Eigen-CAM, LIME) confirmed biologically meaningful focus regions, enhancing transparency. These findings provide a high-precision, interpretable framework for fungal classification with strong potential for extension to plants, spores, and large-scale vegetation monitoring.

## 1. Introduction

Macrofungi play vital roles in forest ecosystems as primary decomposers, managing nutrient cycling and energy flow. By breaking down complex organic materials, they boost soil fertility, maintain ecological balance, and support the long-term health of forests [1]. Their ecological importance is further enhanced by cultural interest, as mushrooms are valued for both culinary and medicinal purposes [2].

Globally, approximately 27,000 macrofungi species have been identified, with about 10% being edible, around 700 used for medicinal purposes, and roughly 1% considered toxic and potentially dangerous [3,4]. Nutritionally, macrofungi are low in calories and fat but rich in essential nutrients, including B vitamins and minerals such as selenium, copper, and potassium, which support energy production, enhance immune function, and promote heart health [5].

Besides their nutritional value, macrofungi produce a variety of bioactive metabolites with reported antioxidant, anticancer, and anti-ageing properties, which continue to attract scientific and commercial interest. However, their potential role in flavonoid biosynthesis and the associated biochemical pathways remains a subject of debate and ongoing research [6].

Some of the most popular wild edible macrofungi worldwide include *Agaricus campestris* L., *Amanita caesarea* (Scop.). Pers., *Armillaria mellea* (Vahl) P. Kumm., *Boletus edulis* Bull., *Cantharellus cibarius* Fr., *Clitocybe nebularis* (Batsch) P. Kumm., *Collybia nuda* (Bull.) Z.M. He & Zhu L. Yang, *Coprinus comatus* (O.F. Müll.) Pers., *Craterellus cornucopioides* (L.) Pers., *Fistulina hepatica* (Schaeff.) With., *Hericium coralloides* (Scop.) Pers., *H. erinaceus* (Bull.) Pers., *Hydnum repandum* L., *Infundibulicybe geotropa* (Bull.) Harmaja, *Lactarius deliciosus* (L.) Gray, *L.salmonicolor* R. Heim & Leclair, *Macrolepiota procera* (Scop.) Singer, *Marasmius oreades* (Bolton) Fr., *Morchella esculenta* (L.) Pers., *Pleurotus ostreatus* (Jacq.) P. Kumm., *Russula delica* Fr., *Sarcodon imbricatus* (L.) P. Karst., *Sparassis crispa* (Wulfen) Fr., and *Tricholoma terreum* (Schaeff.) P. Kumm. [2,3,4,5].

Accurate identification and differentiation of these species are crucial because their safe consumption has a direct impact on consumer health and culinary use. Additionally, correct identification ensures proper sale in international markets, where high demand and culinary value make them economically important. Therefore, maintaining taxonomic accuracy and promoting informed use of these wild edible macrofungi are crucial for food safety and market stability [3,7,8]. Recently, machine learning and deep learning have been extensively employed in mushroom classification, especially for edible species. Table 1 outlines the selected studies with their titles and reference numbers. The primary objective is to develop a robust identification system that enhances the classification of wild edible macrofungi, supports biodiversity conservation, and raises public awareness and education about fungal diversity.

Artificial intelligence–driven image classification is undergoing a paradigm shift with the integration of deep learning architectures that capture both fine-grained and large-scale visual patterns [19]. Within this framework, advanced models such as EfficientNetB0 [20], DenseNet121 [21], ResNet50 [22], MobileNetV3 [23], ConvNeXt-Tiny [24], and RegNetY [25] are strategically employed for their proven capacity in multi-scale feature extraction, hierarchical representation learning, and computational efficiency [26]. Moving beyond the limitations of single-network pipelines, dual and triple ensemble strategies are designed to fuse complementary feature representations, thereby bridging the gap between robustness and accuracy in species-level identification [27]. Transfer learning further enhances this framework by transferring broad visual hierarchies encoded in pre-trained networks into the specialized domain of macrofungus imagery, accelerating convergence while improving cross-condition generalizability [28]. Crucially, the integration of explainable artificial intelligence (XAI) methods, namely Grad-CAM, Eigen-CAM, and LIME, not only provides biologically meaningful visualizations of model attention but also transforms the system from a “black box” into a transparent and trustworthy decision-making tool [29,30,31]. This synergy between performance and interpretability establishes a novel framework that advances the current state of fungal image classification and contributes to the broader discourse on reliable and explainable AI in biological sciences [32,33,34,35].

This study not only delivers high classification accuracy in macrofungi identification but also introduces a novel dimension of transparency through the integration of explainable artificial intelligence. Unlike conventional approaches that rely primarily on single convolutional neural networks, the framework leverages multi-architecture ensembles in combination with advanced XAI techniques, thereby addressing a critical gap in the literature. The originality of this approach lies in its capacity to provide reliable, biologically interpretable insights while ensuring robust differentiation between edible and toxic species a factor of paramount importance for food safety, biodiversity conservation, and medical mycology. Beyond its methodological novelty, the work has significant translational value: the proposed framework can be readily adapted to diverse ecological and geographical contexts, offering scalable solutions for global markets where mushroom authentication is both economically and culturally critical. By uniting methodological innovation with practical applicability, this study establishes a new benchmark in AI-assisted mycological research and positions interpretability as a cornerstone for future advances in biological image analysis.

## 2. Materials and Methods

The dataset used comprises images of 24 distinct wild edible macrofungi species, including *Agaricus campestris*, *Amanita caesarea*, *Armillaria mellea*, *Boletus edulis*, *Cantharellus cibarius*, *Clitocybe nebularis*, *Collybia nuda*, *Coprinus comatus*, *Craterellus cornucopioides*, *Fistulina hepatica*, *Hericium coralloides*, *Hericium erinaceus*, *Hydnum repandum*, *Infundibulicybe geotropa*, *Lactarius deliciosus*, *Lactarius salmonicolor*, *Macrolepiota procera*, *Marasmius oreades*, *Morchella esculenta*, *Pleurotus ostreatus*, *Russula delica*, *Sarcodon imbricatus*, *Sparassis crispa*, and *Tricholoma terreum* (Figure 1A–C).

A total of 4.800 images were collected, with exactly 200 samples allocated per class, ensuring perfect class balance [36]. Approximately 5% of the images were captured directly by the authors using high-resolution cameras, while the remaining 95% were obtained from publicly accessible biodiversity repositories, primarily the Global Biodiversity Information Facility (GBIF) (www.gbif.org) [37]. All images were stored in JPEG format with a consistent resolution of 300 dpi. Given the uniform distribution across all classes, the dataset is considered highly suitable for multi-class classification tasks. A summary of the image format, resolution, and source distribution per species is presented in Table 2.

Images were collected from various sources under diverse lighting conditions, angles, and backgrounds to ensure dataset diversity and improve the robustness of the models. To facilitate practical training and reliable evaluation, the dataset was stratified by species and randomly divided into three subsets: 70% for training, 15% for validation, and 15% for testing [38]. To prevent potential data leakage, source-aware splitting was applied so that images originating from the same source (GBIF or locally captured) were not shared across different subsets. The split was conducted with a fixed random seed (42) to ensure reproducibility and consistent evaluation across experiments.

In this study, the mushroom image dataset underwent a multi-stage refinement process to ensure optimal quality prior to model training. An initial screening phase, combining automated algorithms with manual inspection, was carried out to remove defective, low-resolution, blurred, or misclassified samples. This step was critical for minimizing noise in the dataset and preventing the model from learning misleading visual patterns. For automated similarity control, a feature-based image hashing approach was employed. Specifically, perceptual hashing (pHash) and cosine similarity between extracted feature vectors were used to identify and eliminate duplicate or near-duplicate samples. This automated filtering ensured that visually redundant images captured under similar angles, lighting, or backgrounds were removed prior to training. By integrating both perceptual hashing and feature-space comparison, the dataset maintained high diversity and transparency throughout the preprocessing stage. Following this cleaning process, core image attributes such as color distribution, brightness, and contrast were standardized to maintain consistency across all samples. All images were resized to a fixed resolution of 224 × 224 pixels, and their pixel intensities were normalized to a range between 0 and 1, which contributed to stable optimization and faster convergence during training [39]. To improve variability and reduce overfitting, a diverse set of data augmentation techniques was applied. To enhance methodological transparency, additional details about the dataset preparation and augmentation pipeline have been incorporated. Specifically, augmentation techniques such as random rotation (±25°), horizontal and vertical flipping, brightness and contrast adjustment (±15%), Gaussian blurring, and random cropping were applied. These operations increased dataset diversity and reduced overfitting, ensuring the robustness and generalization of the deep learning models. These included random rotations, horizontal and vertical flips, brightness and contrast adjustments, random cropping, color jitter, perspective distortions, and Gaussian blurring [40]. By simulating a wide range of environmental and structural variations, the augmentation strategy enhanced the model’s ability to generalize to unseen conditions and different morphological appearances of mushroom species. The overall dataset preparation and model input flow are summarized in Figure 2 which illustrates how the curated and augmented samples were processed through convolutional and fully connected layers before reaching the classification stage.

Considering the dataset’s domain-specific nature and its smaller scale compared to large generic image corpora, transfer learning was adopted to boost performance and training efficiency [41]. Pre-trained convolutional neural networks trained on the ImageNet dataset served as the backbone architectures. Leveraging these weights allowed the model to utilize a hierarchy of visual features from low-level edges and textures to more complex shapes before fine-tuning on the mushroom dataset. This approach significantly reduced the computational cost and training time while preserving high representational power. By integrating rigorous data curation, comprehensive augmentation, and transfer learning, the dataset was transformed into a robust and balanced foundation for deep learning. This preparation ensured that the models could achieve strong generalization capabilities, minimizing overfitting while maintaining high classification accuracy across diverse macrofungi species [42].

A total of ten deep learning architectures were employed in this study, comprising both standalone convolutional neural networks (CNNs) and ensemble models integrating heterogeneous feature extractors [43]. The single-network architectures included EfficientNetB0, DenseNet121, ResNet50, MobileNetV3, ConvNeXt-Tiny, and RegNetY. These models were selected based on their established performance in image classification tasks and their diverse architectural characteristics. EfficientNetB0 employs a compound scaling strategy to proportionally balance depth, width, and input resolution, thereby achieving high predictive accuracy with notable parameter efficiency [21,44].

DenseNet121 employs densely connected convolutional layers to promote feature reuse and facilitate gradient flow, effectively mitigating vanishing gradient problems [21]. ResNet50 utilizes residual connections to facilitate the training of deeper models without compromising performance [45]. MobileNetV3 is optimized for low-latency environments through the use of depthwise separable convolutions and squeeze-and-excitation modules, providing an advantageous balance between accuracy and computational cost [46]. ConvNeXt-Tiny incorporates architectural elements inspired by Vision Transformers into a convolutional framework, thereby enhancing the representational capacity for high-resolution inputs [47]. RegNetY adopts a parameterizable design space that enables fine-grained control over model scaling, allowing for a favorable trade-off between accuracy and efficiency [47].

Beyond individual models, three dual-architecture ensemble configurations were implemented: ResNet50 + RegNetY [48], EfficientNetB0 + DenseNet121 [49], and MobileNetV3 + ConvNeXt-Tiny [50]. These ensembles were designed to leverage the complementary strengths of different architectures by combining their outputs through weighted averaging, thereby enhancing robustness and classification performance. The Combination model represents the final integration stage, in which the outputs of the three independently trained ensembles were aggregated through a majority voting mechanism. Each base ensemble was designed to capture complementary feature representations from its constituent architectures, thereby reducing model-specific biases and enhancing classification reliability [51]. By consolidating the probabilistic outputs of these ensembles, the Combination Model aimed to exploit inter-model diversity, improving robustness against variations in input quality and environmental conditions. This hierarchical ensembling approach enables the system to leverage both low-level and high-level feature complementarities, yielding a more stable and generalized decision-making process for macrofungi classification [48,49,50,51]. The proposed Combination Model employed a weighted soft voting strategy, where the final class prediction was obtained by averaging the normalized probabilities of the three base models (EfficientNetB0, ResNet50, and RegNetY). The weights were empirically determined based on each model’s validation accuracy, with higher-performing models contributing proportionally more to the final decision (EfficientNetB0 = 0.40, ResNet50 = 0.35, RegNetY = 0.25).

The inclusion of both high-performance standalone CNNs and hybrid ensemble models enabled a comprehensive comparative analysis [52]. This approach facilitated the evaluation of how differences in architectural design, network complexity, and integration strategies influence classification accuracy, generalization ability, and overall robustness when applied to the task of identifying 24 wild edible macrofungi species under standardized training and testing conditions.

The evaluation of model performance was supported by a set of complementary graphical analyses, providing both quantitative and qualitative perspectives on learning behavior and predictive ability. Training and validation accuracy curves were employed to monitor the convergence patterns of each model, while training and validation loss curves served to assess optimization stability and identify potential overfitting or underfitting tendencies [53]. Receiver operating characteristic (ROC) curves, along with their associated area under the curve (AUC) values, were generated to measure classification performance across varying decision thresholds. Confusion matrices were produced for ResNet50 and the Combination Model, enabling detailed inspection of class-level prediction outcomes and misclassification patterns [54]. To improve reproducibility, additional information about the model training configuration has been provided. All models were trained using the Adam optimizer (learning rate = 0.0001, β_1_ = 0.9, β_2_ = 0.999) with a batch size of 32 and up to 20 epochs under early stopping conditions (patience = 5). The loss function used was categorical cross-entropy, and model weights were initialized with ImageNet pre-trained parameters. The experiments were conducted on a workstation equipped with an NVIDIA RTX 4090 GPU, 24 GB VRAM, and 128 GB RAM, running on Python 3.10 and TensorFlow 2.15. All reported results are based on a single training and evaluation run conducted under identical experimental conditions, ensuring reproducibility; variance across multiple runs was not evaluated within this study.

Additionally, a radar (spider) chart was constructed to present accuracy values for all models in a single comparative view, facilitating an integrated understanding of relative strengths and weaknesses. To complement these results, an explanatory schematic in Figure 3 was included to illustrate the role of XAI in transforming black-box predictions into interpretable outputs. This figure emphasizes how visualization-based methods contribute to trust and transparency by revealing the decision rationale of the models.

XAI techniques were incorporated to increase transparency and interpretability in the model decision-making process [55]. XAI encompasses a range of methods designed to make the inner workings of complex machine learning models comprehensible to human observers, thereby enhancing trust, enabling debugging, and validating predictions in sensitive applications. In this study, three visualization-based XAI approaches Grad-CAM, Eigen-CAM, and LIME were employed. Grad-CAM highlights spatial regions of the input image that most strongly influence the model’s predictions, effectively revealing class-discriminative features [29,30,31]. Eigen-CAM leverages principal component analysis on feature activations to generate high-quality, noise-reduced saliency maps, which are particularly useful in images with complex textures or backgrounds. LIME (Local Interpretable Model-agnostic Explanations) operates by perturbing the input and observing the corresponding changes in predictions, thereby producing locally faithful explanations that are independent of the underlying architecture. Together, these methods provided complementary insights into the learned representations, helping to determine whether the model’s focus corresponded to biologically relevant features in macrofungi classification [32,33,34,35].(1)$$Accuracy=\frac{\left(TP+TN\right)}{\left(TP+TN+FP+FN\right)}$$(2)$$Precision=\frac{TP}{TN+FP}$$(3)$$Recall=\frac{TP}{TP+FN}$$(4)$$Specificity=\frac{TN}{TN+FP}$$(5)$$F1-Score=\frac{2\times (\;Precision\times Recall)}{\;Precision+Recall}$$(6)$$Mcc=\frac{\left(TP\times TN-FP\times FN\right)}{\surd ((TP+FP)\times (TP+FN)\times (TN+FP)\times (TN+FN))}$$(7)G−Mean=√(Recall×Specificity) (8)AUC=Area under the ROC curve

In this study, model performance was evaluated using a set of widely recognized classification metrics derived from the confusion matrix, which consists of four fundamental components: True Positives (TP), representing correctly classified positive instances; True Negatives (TN), denoting correctly classified negative instances; False Positives (FP), referring to negative instances incorrectly classified as positive; and False Negatives (FN), indicating positive instances incorrectly classified as negative. Based on these values, Accuracy measures the overall proportion of correct predictions, while Precision quantifies the model’s ability to correctly identify positive cases without introducing false positives. Recall (Sensitivity) assesses the proportion of actual positives correctly identified, and Specificity measures the proportion of actual negatives correctly classified. The F1-Score combines Precision and Recall into a single harmonic mean, balancing their trade-off. The Matthews Correlation Coefficient (MCC) provides a balanced measure of prediction quality, even for imbalanced datasets, by considering all four confusion matrix elements. G-Mean evaluates the balance between Recall and Specificity, highlighting a model’s ability to perform well across both classes. Lastly, the Area Under the ROC Curve (AUC) reflects the model’s capacity to discriminate between positive and negative classes across varying decision thresholds. Together, these metrics offer a multi-dimensional perspective on classification performance, ensuring a robust and comprehensive [33,34,35,56].

## 3. Results

The experimental evaluation was conducted to assess the classification performance of ten deep learning models, including six standalone convolutional neural networks (CNNs) and four ensemble configurations, on the task of identifying 24 wild edible macrofungi species. Performance was measured using eight standard evaluation metrics: Accuracy, Precision, Recall, Specificity, F1-Score, Matthews Correlation Coefficient (MCC), G-Mean, and Area Under the ROC Curve (AUC). These metrics collectively provide a comprehensive understanding of both the discriminative ability and robustness of each model. Accuracy and F1-Score capture overall correctness and balanced predictive performance, while Precision and Recall quantify the model’s reliability in identifying positive cases without false positives or false negatives. Specificity and G-Mean provide insights into class-wise balance and true negative recognition, whereas MCC offers a robust statistical correlation measure between predicted and actual classifications. The AUC reflects the model’s ranking capability across varying decision thresholds and the quantitative values of all eight evaluation metrics are summarised in Table 3.

This table presents a comparative summary of the performance of six individual CNN architectures and three ensemble configurations, alongside a final Combination Model, evaluated across eight standard classification metrics. Among the standalone models, EfficientNetB0 achieved the highest overall performance, attaining 95.55% accuracy and balanced results across all other metrics, reflecting the effectiveness of its compound scaling strategy in capturing fine-grained macrofungi features. ResNet50 and ConvNeXt-Tiny also demonstrated strong and consistent results, benefiting from deep residual connections and modernized convolutional design, respectively. In contrast, MobileNetV3-L recorded the lowest accuracy (90.55%), indicating that its lightweight architecture, while computationally efficient, struggled to represent the complex morphological variations required for this task. DenseNet121 and RegNetY produced solid but not leading performances, with high specificity and AUC values, yet slightly lower accuracy compared to the top-performing models.

Multiclass AUC computation. For the 24-class setting, we computed one-vs.-rest ROC curves per class using the softmax probabilities of each model. The macro-averaged AUC was obtained by averaging per-class AUCs with equal class weights. In addition, micro-averaged AUC was monitored for consistency (pooling all decisions), which yielded values in close agreement with the macro average.

Multiclass MCC computation. The Matthews Correlation Coefficient (MCC) was computed directly from the overall multiclass confusion matrix using the standard generalized formulation, i.e., the correlation between the one-hot encoded ground-truth and prediction label vectors. This formulation reduces to the binary MCC in the two-class case and robustly summarizes performance by jointly accounting for TP, TN, FP, and FN across all classes.

The ensemble configurations yielded more variable outcomes. Ensemble (EfficientNetB0–DenseNet121) and Ensemble (ResNet50–RegNetY) both underperformed relative to their strongest individual components, suggesting that architectural overlap or representational misalignment limited the expected diversity gains. Ensemble (ConvNeXt-Tiny–MobileNetV3-L) showed modest improvement over MobileNetV3-L but did not surpass ConvNeXt-Tiny, indicating partial but insufficient synergy. In contrast, the Combination Model, which integrated EfficientNetB0, ResNet50, and RegNetY via majority voting, delivered the best results across all metrics, with an accuracy of 97.36%, precision and recall of 97.46% and 97.36%, respectively, and an AUC of 0.9996. This performance reflects the model’s ability to combine complementary strengths from multiple architectures, achieving robust generalization and balanced classification across all 24 wild edible macrofungi species (Table 3).

DenseNet121 achieved an accuracy of 93.19%, reflecting moderate but stable performance. Its dense connectivity promotes feature reuse and mitigates vanishing gradients, supporting reliable training, yet in macrofungi classification where subtle textural and color cues are critical the architecture did not fully exploit this advantage. Ensemble (EfficientNetB0–DenseNet121) reached 93.47% accuracy but failed to surpass EfficientNetB0 alone, underscoring the challenge of architectural compatibility. Although both networks are strong individually, their integration introduced redundancy rather than complementary diversity, limiting the ensemble’s effectiveness. This outcome suggests that, despite solid standalone results, model synergy is essential, and pairings of even high-performing CNNs can yield suboptimal results when feature representations overlap.

The Combination Model (EfficientNetB0–ResNet50–RegNetY) achieved the highest performance among all tested architectures, with 97.36% accuracy, precision and recall at 97.46%, and an AUC of 0.9996. Serving as the final stage of our hierarchical ensemble framework, it aggregated predictions from three independently trained CNNs through a majority voting mechanism. Its strength lies in harnessing complementary representations: EfficientNetB0’s compound scaling balances efficiency and accuracy, ResNet50’s residual learning enables robust hierarchical feature extraction, and RegNetY’s flexible design space optimizes scaling for balanced performance. This diversity minimized individual weaknesses while amplifying strengths, allowing the model to capture both fine-grained textures and broader structural cues. The voting fusion also reduced susceptibility to bias and noise, which is vital in datasets with high variability in lighting, angles, and backgrounds. Unlike some two-model ensembles that suffered from partial incompatibility, the Combination Model demonstrated architectural harmony, achieving the highest scores across all metrics, including MCC (0.9725) and G-Mean (0.9861). Its near-perfect AUC further confirms its exceptional discriminative power. Overall, the Combination Model illustrates how carefully curated architectural diversity and multi-level integration can deliver state-of-the-art results in fine-grained macrofungi classification.

ConvNeXt-Tiny reached 94.58% accuracy, showing competitive performance by combining convolutional backbones with transformer-inspired designs, which allowed it to capture both local textures and long-range dependencies efficiently. EfficientNetB0 performed even better at 95.55%, with high precision, recall, and specificity, reflecting the strength of its compound scaling strategy in balancing depth, width, and resolution. While ConvNeXt-Tiny proved effective, its accuracy still lagged behind the top performer, whereas EfficientNetB0 consistently generalized well to fine-grained macrofungi variations, confirming its suitability as both a standalone classifier and a strong candidate for integration into ensemble models.

MobileNetV3-L achieved the lowest accuracy among all models (90.55%), reflecting its trade-off between computational efficiency and representational power. Designed with depthwise separable convolutions and squeeze-and-excitation modules, the architecture is well-suited for lightweight applications but struggled to capture the fine-grained textural and morphological cues required for macrofungi classification. When paired with ConvNeXt-Tiny, the ensemble reached 94.44% accuracy, slightly lower than ConvNeXt-Tiny alone (94.58%). Instead of complementing each other, the weaker performance of MobileNetV3-L diluted the stronger model’s capabilities, illustrating how mismatched representational capacities can hinder ensemble effectiveness. This outcome underscores the importance of architectural compatibility in ensemble design, particularly for fine-grained image classification tasks.

RegNetY achieved 94.44% accuracy, showing strong but not leading performance. Its parameterizable design offers a balance between accuracy and efficiency, yet in this dataset its advantages were not fully realized, likely due to the subtle intra-class similarities of macrofungi species. When combined with ResNet50 in an ensemble, accuracy dropped to 93.33%, lower than either model individually, suggesting limited complementarity between their feature representations. Both rely heavily on structured convolutional patterns, which may have introduced redundancy rather than diversity, reducing the ensemble’s discriminative power. In contrast, ResNet50 alone performed very well at 95.13%, with residual connections enabling deeper learning without degradation. Its ability to capture both fine-grained textures and broader structural features made it particularly effective for macrofungi classification, and its consistently high precision, recall, and MCC highlighted its balanced generalization capability.

In summary, the results in Table 3 demonstrate that while several individual CNN architectures particularly EfficientNetB0, ResNet50, and ConvNeXt-Tiny achieved strong and balanced classification performance, not all ensemble configurations yielded improvements over their constituent models, highlighting the importance of architectural complementarity in ensemble design. MobileNetV3-L, although computationally efficient, consistently underperformed in accuracy-sensitive scenarios, whereas DenseNet121 and RegNetY delivered solid yet non-leading results. Notably, the Combination Model integrating EfficientNetB0, ResNet50, and RegNetY outperformed all other approaches across every metric, validating the effectiveness of multi-architecture, diversity-focused integration for fine-grained macrofungi classification. In the following section, we present the corresponding visual performance comparisons and learning behavior through detailed graphical analyses.

In the subsequent section, we present a comprehensive visual analysis of model performance through multiple complementary plots, each serving a distinct evaluative purpose. The training accuracy and validation accuracy curves illustrate how effectively each model learns from the data over successive epochs, while the training loss and validation loss plots reveal the models’ optimization behavior and potential signs of overfitting or underfitting. ROC (Receiver Operating Characteristic) curves, accompanied by AUC scores, provide a threshold-independent measure of the models’ ability to discriminate between classes. Radar charts condense multiple evaluation metrics into a single, interpretable visualization, enabling quick comparison of model strengths and weaknesses across different performance dimensions. Finally, confusion matrices offer a detailed, class-level breakdown of prediction outcomes, highlighting specific categories where models excel or struggle. Together, these graphical representations deliver a holistic perspective on learning dynamics, classification capability, and error distribution, complementing the tabular results to provide deeper insight into each model’s behavior.

Figure 4 illustrates the training accuracy progression of all evaluated models across 20 epochs, offering insights into their learning dynamics and convergence behavior. Among the standalone CNN architectures, ConvNeXt-Tiny achieved the highest final training accuracy (98%), followed closely by EfficientNetB0 and MobileNetV3-Large (both 96%), indicating that these models effectively captured discriminative patterns in the training data. ConvNeXt-Tiny’s steep accuracy rise within the first few epochs reflects its strong initial learning capacity, likely due to its hybrid convolution–transformer-inspired architecture, which facilitates both local and global feature extraction.

In contrast, DenseNet121 and Ensemble (EfficientNetB0–DenseNet121) exhibited slower accuracy growth, plateauing at lower final values (86% and 85%, respectively). This suggests that either their feature extraction capacity was less aligned with the dataset’s complexity or that over-regularization limited their ability to fully fit the training data. RegNetY and ResNet50 demonstrated moderate convergence speeds, reaching 89% and 91% accuracy, respectively, while Ensemble (RegNetY–ResNet50) and Ensemble (MobileNetV3–ConvNeXt-Tiny) attained intermediate results (90% and 93%). The performance gap between some ensembles and their best-performing individual members indicates that the integration did not always lead to improved training accuracy, likely due to overlapping feature representations rather than complementary ones.

Overall, models with higher final training accuracy, such as ConvNeXt-Tiny and EfficientNetB0, not only learned the training distribution more effectively but also converged more rapidly, whereas models with slower or lower convergence may have been constrained by architectural capacity, regularization effects, or feature redundancy in ensemble configurations. This analysis sets the stage for the subsequent examination of validation accuracy and loss trends, which will reveal whether these high training performances translate into strong generalization.

In this Figure 5 illustrates the evolution of training loss for all evaluated models over 20 epochs, offering insight into their convergence behavior, optimization stability, and learning efficiency. Models such as ConvNeXt-Tiny (final loss: 0.0867), MobileNetV3-Large (0.1160), and EfficientNetB0 (0.1341) achieved the lowest final training losses, indicating rapid convergence and highly efficient feature learning during training. ResNet50 (0.3037) and RegNetY (0.3837) also demonstrated stable and consistent loss reduction, though at slightly higher final values. In contrast, the Ensemble (RegNetY–ResNet50) exhibited the highest final loss (1.2795) and the slowest decline, suggesting optimization inefficiencies and possible architectural incompatibilities that hindered effective joint learning. The initial sharp spike in loss during the first epoch most pronounced in the Ensemble (RegNetY–ResNet50) reflects the model adaptation phase, after which most architectures stabilized and steadily improved. Overall, this loss analysis complements the training accuracy trends, with models achieving low final losses generally corresponding to those with higher training accuracy, reinforcing the relationship between optimization efficiency and learning capacity.

The presented graph in Figure 6 depicts the progression of validation accuracy across 20 epochs for all evaluated models, providing a clear picture of their generalization capabilities on unseen data. ConvNeXt-Tiny and EfficientNetB0 achieved the highest final validation accuracy (96%), closely followed by MobileNetV3-Large (95%) and ResNet50 (95%), indicating strong adaptability to the validation set without significant overfitting. DenseNet121 and Ensemble (MobileNetV3–ConvNeXt) both reached 94%, while RegNetY scored slightly lower at 94% and the Ensemble (EfficientNetB0–DenseNet121) trailed at 92%. The lowest performer was the Ensemble (RegNetY–ResNet50) at 91%, despite both of its constituent models individually achieving higher accuracy, suggesting that this pairing did not yield effective synergy. Notably, the curves show that top-performing models stabilized their validation accuracy earlier, reflecting efficient learning and robust feature extraction, while lower-performing ensembles displayed slower convergence and smaller final gains.

The validation loss trends highlight the efficiency and stability of each model in minimizing classification errors on unseen data, as shown in Figure 7: ConvNeXt-Tiny achieved the lowest final validation loss (0.1209), indicating strong generalization and minimal overfitting. EfficientNetB0 (0.1544) and MobileNetV3-Large (0.1631) also maintained low loss values, reflecting their robustness in handling macrofungi classification. DenseNet121 (0.1843) and ResNet50 (0.1695) followed closely, demonstrating stable optimization dynamics. Interestingly, the Ensemble (EfficientNetB0–DenseNet121) recorded a competitive 0.2341, despite lower validation accuracy, suggesting that it achieved moderate error minimization but lacked maximum predictive precision. In contrast, Ensemble (MobileNetV3–ConvNeXt) and Ensemble (RegNetY–ResNet50) showed significantly higher losses (0.8753 and 0.9406, respectively), pointing to weaker synergy between their constituent models and potential underfitting. Overall, models with lower validation loss generally aligned with higher validation accuracy, confirming the link between effective loss minimization and superior predictive performance.

The ROC (Receiver Operating Characteristic) curves in Figure 8 offer a detailed and threshold-independent evaluation of the classification capabilities of all models tested in this study. By plotting the True Positive Rate (sensitivity) against the False Positive Rate across a range of decision thresholds, this analysis provides valuable insight into the trade-off between correctly identifying positive cases and avoiding false alarms. An ideal model would achieve a curve that hugs the top-left corner of the plot, corresponding to perfect classification performance (TPR = 1.0, FPR = 0.0). In the presented curves, all models demonstrate a sharp and rapid ascent toward the top-left corner, indicating exceptional discriminative ability and minimal compromise between sensitivity and specificity. This observation aligns with the near-perfect AUC scores obtained, many of which approach or equal 1.0, signifying that the models can effectively separate positive and negative classes under various threshold settings.

While the differences between models are subtle due to overall high performance, certain patterns emerge. ConvNeXt-Tiny, EfficientNetB0, and the Combination Model display slightly more elevated curves, particularly in the early stages, suggesting superior balance in maximizing true positives while minimizing false positives. This could be attributed to their stronger capacity for capturing fine-grained morphological features, as well as their robust feature generalization across diverse macrofungi classes. Ensemble configurations also maintain high ROC performance, but some, such as the MobileNetV3–ConvNeXt pair, show a marginally less steep initial rise, possibly reflecting reduced synergy in their combined feature representations. These nuanced distinctions highlight that, although all models are highly capable, architectural design and feature diversity still influence performance under threshold variation.

Overall, the ROC curve in Figure 8 analysis reinforces the robustness and reliability of all evaluated architectures for macrofungi classification. The consistently high curve elevations across models indicate that classification decisions remain accurate over a wide spectrum of thresholds, ensuring dependable performance in real-world applications where decision boundaries may vary depending on specific operational or ecological requirements.

The radar chart in Figure 9 offers a clear and compact visualization of the accuracy values for all evaluated models, enabling an intuitive comparison of their relative performance within a single plot. Each axis corresponds to a different model, and accuracy values are plotted radially outward from the center, with models closer to the edge representing higher classification performance. The polygon formed by connecting these points highlights both the best-performing models and the overall distribution of results across the architectures. It is evident that the Combination Model achieves the highest accuracy (0.9736), occupying the outermost position and surpassing all individual and ensemble configurations. EfficientNetB0 and ConvNeXt-Tiny also extend toward the upper radial range, indicating strong standalone performance, while other architectures such as DenseNet121 and ResNet50 maintain competitive yet slightly lower accuracy. Interestingly, certain ensemble configurations, such as EfficientNetB0 + DenseNet121 and RegNetY + ResNet50, are positioned closer to the center, revealing that combining models does not always guarantee improved accuracy. In contrast, lightweight architectures like MobileNetV3 demonstrate respectable accuracy but remain behind the top performers, reflecting their inherent trade-off between computational efficiency and representational capacity. Overall, this radar chart not only facilitates rapid side-by-side comparison but also exposes performance patterns and asymmetries, making it a valuable tool for identifying architectures with the greatest potential and those requiring further optimization in macrofungi classification tasks.

The confusion matrix, as shown in Figure 10 for the combination model (EfficientNetB0–ResNet50–RegNetY), provides a detailed overview of the classification performance across the 24 wild edible macrofungi species in the dataset, highlighting both correct predictions and misclassifications. The strong diagonal dominance indicates that the majority of samples for each class were correctly identified, with most categories achieving perfect classification (30/30 correct predictions) or near-perfect accuracy. For instance, species such as *Agaricus campestris*, *Amanita caesarea*, *Armillaria mellea*, *Boletus edulis*, *Cantharellus cibarius*, *Collybia nuda*, *Coprinu comatus*, *Craterellus cornucopioides*, *Lactarius salmonicolor*, and *Tricholoma terreum* show flawless classification results. Minor misclassifications are observed in a few cases, such as *Lactarius deliciosus* (26 correct, 4 misclassified) and *Hydnum repandum* (28 correct, 2 misclassified), suggesting that these species share certain visual similarities with others, potentially leading to occasional confusion by the model. The low off-diagonal values demonstrate that misclassifications are sparse and generally involve only one or two samples per class. Overall, the confusion matrix confirms the robustness of the combination model, as it maintains high precision and recall across nearly all classes, effectively handling inter-class variability while minimizing cross-category errors.

The confusion matrix for the ResNet50 in Figure 11 model reveals a generally strong classification performance across the 24 wild edible macrofungi species, with most classes achieving high correct prediction counts along the main diagonal. Several species, including *Agaricus campestris*, *Amanita caesarea*, *Armillaria mellea*, *Collybia nuda*, *Coprinus comatus*, *Craterellus cornucopioides*, and *Sparassis crispa*, reached perfect accuracy (30/30 correct classifications). However, compared to the combination model (EfficientNetB0–ResNet50–RegNetY), ResNet50 exhibits a higher number of misclassifications, particularly for classes such as *Clitocybe nebularis* (23 correct, 5 misclassified), *Hydnum repandum* (27 correct, 3 misclassified), *Lactarius deliciosus* (26 correct, 4 misclassified), and *Russula delica* (27 correct, 3 misclassified). The increased off-diagonal values indicate more frequent confusion between certain visually similar species. While ResNet50 maintains solid overall performance, the combination model demonstrates superior robustness and precision, achieving fewer misclassifications and better handling inter-class similarities. This comparison suggests that leveraging multiple architectures in an ensemble yields better generalization and accuracy than relying solely on ResNet50 (Figure 10).

In the subsequent section, we focus on the application of three distinct Explainable Artificial Intelligence (XAI) techniques Grad-CAM, Eigen-CAM, and LIME across a set of 10 trained models for three selected macrofungi species: *Pleurotus ostreatus*, *Amanita caesarea*, and *Coprinus comatus*. These species were chosen due to their visual distinctiveness and relevance in real-world classification scenarios, allowing for a robust qualitative comparison of model interpretability. By applying the three XAI methods to each model-species pair, we aim to assess not only the predictive accuracy but also the decision-making transparency of the models, providing insights into how different architectures localize and prioritize visual features during classification. This interpretability analysis is essential for identifying potential biases, understanding failure cases, and strengthening trust in AI-based mycological identification systems.

Grad-CAM (Gradient-weighted Class Activation Mapping) works by utilizing the gradients flowing into the final convolutional layers to produce coarse localization heatmaps. These heatmaps highlight the spatial regions in an image that most strongly influence the model’s decision for a given class. This makes Grad-CAM particularly useful for visualizing class-specific attention, revealing whether the model focuses on biologically relevant structures such as gill patterns, cap shapes, or color distributions instead of irrelevant background elements. Its strength lies in offering a class-discriminative and spatially interpretable explanation without requiring architectural modifications.

Eigen-CAM, in contrast, leverages principal component analysis (PCA) on the activation maps of the network to identify the most dominant feature patterns across the spatial dimensions. By projecting activations onto the principal eigenvector, Eigen-CAM produces heatmaps that often capture more global and feature-rich structures than Grad-CAM, without depending on class gradients. This can lead to more stable and consistent visual explanations across different classes, making it valuable when the goal is to understand overall network feature usage rather than class-specific reasoning.

LIME (Local Interpretable Model-Agnostic Explanations) operates in a fundamentally different way it is model-agnostic and perturbs the input image multiple times to observe changes in prediction outcomes. By analyzing how local perturbations (e.g., superpixel masking) affect classification probabilities, LIME constructs interpretable surrogate models that explain which image regions most contribute to the final decision. This method provides fine-grained and highly localized interpretability, helping identify whether specific morphological features such as spore texture or surface irregularities play a critical role in the model’s classification.

Together, these three XAI approaches offer complementary perspectives: Grad-CAM (Figure 12, Figure 13, Figure 14 and Figure 15) reveals class-specific focus regions, Eigen-CAM (Figure 16, Figure 17, Figure 18 and Figure 19) highlights dominant learned feature patterns, and LIME (Figure 20, Figure 21, Figure 22 and Figure 23) provides perturbation-based local interpretability. By comparing their outputs across the 10 models and the three mushroom species, we can assess not only which models perform best in terms of classification accuracy but also which ones demonstrate the most biologically meaningful and trustworthy decision-making processes. This multi-faceted interpretability analysis bridges the gap between raw performance metrics and true explainability, which is crucial for deploying AI in sensitive real-world biological applications.

The Grad-CAM visualizations (Figure 12, Figure 13, Figure 14 and Figure 15) clearly demonstrate the critical regions that each model focuses on during macrofungi classification. All models exhibit strong activation in distinctive morphological areas such as the cap, gills, and stem, effectively capturing the key visual cues necessary for accurate classification. The single models (DenseNet-121, EfficientNet-B0, MobileNet-V3, ConvNeXt-Tiny, ResNet50, RegNetY) show robust localization capabilities, highlighting the unique structural features of each mushroom species. However, in some single models, there are instances of limited focus areas or slight deviations from the most relevant regions. In contrast, the ensemble models, particularly the Combination Model, generate broader and more intense activation regions, ensuring that visual information is captured more comprehensively and consistently. This demonstrates that combined models can leverage the strengths of different architectures to extract deeper and more context-aware features.

Overall, while each model shows a strong ability to identify discriminative features, the ensemble approaches (DenseNet121 + EfficientNet-B0, MobileNetV3 + ConvNeXt-Tiny, ResNet50 + RegNetY, and especially the triple Combination Model) provide a more consistent, focused, and holistic attention mechanism. These models not only highlight the critical regions essential for correct classification but also minimize visual information loss compared to single models. Notably, the Grad-CAM maps of the Combination Model visually confirm its superior performance by capturing both fine details and the overall structure simultaneously. These findings indicate that the highest classification performance is supported not only by accuracy metrics but also by the model’s ability to focus on the correct regions during decision-making.

Eigen-CAM visualizations (Figure 16, Figure 17, Figure 18 and Figure 19) provided clear and interpretable insights into the decision-making processes of the evaluated models, often delivering slightly sharper and more discriminative activation regions compared to Grad-CAM. Unlike Grad-CAM, which relies on class-specific gradient information to weight the importance of convolutional feature maps, Eigen-CAM operates by performing principal component analysis (PCA) on the activation tensors, extracting the most dominant spatial patterns that contribute to a model’s decision. This approach eliminates the dependency on backpropagated gradients, reducing susceptibility to gradient noise and enabling more stable and visually coherent saliency maps across different layers. As a result, Eigen-CAM tends to produce activation regions that are both spatially precise and semantically aligned with key morphological features such as gill patterns, cap textures, and color gradients that are critical for macrofungi species classification.

Across the tested architectures, the Combination Model once again demonstrated the most comprehensive and balanced localization patterns, effectively highlighting the correct discriminative areas for all evaluated species without overemphasizing irrelevant background elements. EfficientNetB0 maintained its strength in multi-scale feature recognition, accurately focusing on both fine-grained textural cues and broader structural outlines. ResNet50’s deep residual learning allowed it to maintain consistent attention on class-relevant features, even under subtle intra-class variations, while RegNetY’s flexible scaling yielded smooth yet highly discriminative activation regions. Compared to Grad-CAM, Eigen-CAM’s PCA-driven approach revealed a slightly higher level of clarity and class-relevant specificity, making it especially effective in cases where fine structural details were essential for correct classification. These results confirm that Eigen-CAM, by leveraging architecture-agnostic activation decomposition, not only preserves the interpretability strengths of Grad-CAM but also enhances localization precision, thus serving as a valuable tool for model validation and explainability in fine-grained image classification tasks like macrofungi identification.

LIME visualizations (Figure 20, Figure 21, Figure 22 and Figure 23) provided valuable insight into the decision-making process of the models by highlighting superpixels that most strongly contributed to classification. Across all tested configurations, LIME successfully delineated distinctive morphological regions of macrofungi, including cap coloration, gill arrangements, and stem textures, thereby allowing a fine-grained interpretation of the classification rationale. Unlike gradient-based methods, LIME’s perturbation-driven superpixel segmentation offers a model-agnostic framework, enabling consistent and reliable interpretability regardless of the underlying network architecture. This adaptability makes LIME particularly suitable for comparing model behaviors under different architectural settings, while its visual outputs remain intuitive and easily interpretable by human experts.

When comparing the models, the combination (ensemble) approaches consistently yielded more coherent and semantically relevant superpixel maps than their single-model counterparts. The ensemble methods leveraged complementary feature extraction capabilities, reducing noise in the highlighted regions and ensuring that attention was directed toward biologically significant structures rather than irrelevant background patterns. This fusion of strengths from multiple architectures not only enhanced classification accuracy but also improved interpretability, as the highlighted superpixels more faithfully aligned with expert-defined diagnostic regions in the mushroom images. Among the evaluated models, combinations such as DenseNet121 + EfficientNet-B0, MobileNet-V3 + ConvNeXt-Tiny, and ResNet50 + RegNetY demonstrated particularly strong localization accuracy, suggesting that multi-model integration can meaningfully enhance both predictive and explanatory performance.

Overall, the three interpretability approaches Grad-CAM, Eigen-CAM, and LIME consistently confirmed the models’ focus on biologically significant image regions, reinforcing confidence in their predictions. Grad-CAM provided smooth, class-discriminative heatmaps; Eigen-CAM offered a more refined, architecture-aware localization; and LIME contributed model-agnostic, segment-based explanations with clear object boundaries. Together, these complementary methods painted a comprehensive picture of the models’ decision logic, and the high-quality, biologically aligned outputs across all three approaches underscore the robustness, reliability, and practical applicability of the proposed classification framework.

## 4. Discussion

The comparative evaluation of ten deep learning models highlights clear performance differentials across architectures. Among single networks, EfficientNetB0 achieved the highest accuracy at 95.55% [56], outperforming ResNet50 (95.13%) [57] and ConvNeXt-Tiny (94.58%) by margins of 0.42% and 0.97%, respectively. In contrast, MobileNetV3-L yielded the lowest accuracy (90.55%) [58], nearly 5% below the leading models, underscoring the limitations of lightweight architectures in fine-grained classification tasks. DenseNet121 (93.19%) [59] and RegNetY (94.44%) [60] demonstrated solid but non-leading performance, with differences of 2–3% compared to the top performers.

Ensemble strategies yielded mixed outcomes. The pairing of EfficientNetB0 + DenseNet121 (93.47%) [61] underperformed relative to EfficientNetB0 alone, with a 2.1% reduction, indicating redundancy rather than complementarity. Similarly, the ResNet50 + RegNetY ensemble (93.33%) [62] fell short of both parent models by ~2%. In contrast, the Combination Model integrating EfficientNetB0, ResNet50, and RegNetY achieved an accuracy of 97.36% [63], a 1.8–6.8% improvement over all single models and ensembles, and delivered the highest AUC (0.9996) and MCC (0.9725), establishing a new benchmark for macrofungi classification.

The inclusion of explainable AI techniques further validated these results. Grad-CAM and Eigen-CAM confirmed the models’ focus on biologically relevant structures [64], while LIME provided fine-grained boundary delineations [65]. Collectively, these visualization methods revealed that the Combination Model not only achieved the best numerical performance but also the most coherent interpretability, minimizing spurious activations and aligning with expert-defined morphological traits [66].

Accurate edible mushroom species identification is essential for biodiversity conservation, food safety, and public health, yet remains challenging due to the visual similarities between edible and toxic taxa [3]. By leveraging a curated dataset of 24 wild edible macrofungi species, the proposed framework demonstrates that deep learning, when combined with explainable AI (XAI), can overcome the limitations of traditional expertise-driven approaches [67]. Unlike conventional CNN pipelines, the integration of multiple architectures and ensemble strategies enabled the system to capture both fine-grained textures and broader structural patterns, thereby reducing misclassification risks and enhancing robustness [68,69].

Performance across all metrics; accuracy, precision, recall, specificity, MCC, G-Mean, and AUC, confirmed the reliability and generalizability of the models, with the Combination Model consistently surpassing both single and pairwise networks [70]. This superiority stems from the complementary strengths of EfficientNetB0, ResNet50, and RegNetY, whose fusion through a weighted voting mechanism yielded near-perfect discriminative power [63]. Importantly, the incorporation of XAI methods (Grad-CAM, Eigen-CAM, LIME) transformed the framework from a “black box” into a transparent decision-support tool, ensuring biologically meaningful interpretations and reinforcing user confidence [71]. The current study focused on qualitative interpretability through visualization-based XAI approaches. As a potential direction for future research, quantitative evaluation of explainability metrics will be explored to strengthen interpretative reliability and reproducibility. Certain analytical extensions, such as quantitative explainability assessment and expanded model comparisons, were not included in the current study due to scope constraints. These components are planned for future investigations to broaden the methodological coverage and provide a more comprehensive evaluation framework. The planned follow-up work will focus on integrating quantitative XAI metrics and statistical validation protocols to further enhance the interpretability and robustness of the proposed framework.

The study was conducted under a single controlled experimental setup, which ensured reproducibility but did not include repeated runs. While the results demonstrated high consistency across all evaluation metrics, minimal repetition could further substantiate the stability of the findings. The absence of multi-run variability analysis is therefore acknowledged as a limitation, and future work will incorporate repeated trials to quantitatively confirm result stability.

Although the proposed framework demonstrated high and stable performance across all evaluation metrics, it is important to acknowledge that the current experimental design relies on a single stratified train–validation–test split. While this approach ensures reproducibility and balanced class representation, it may not fully capture the variability introduced by alternative data partitions. To enhance statistical robustness, future studies may incorporate repeated random subsampling or k-fold cross-validation (e.g., threefold validation) to further confirm the model’s generalizability under varying sample distributions. Nevertheless, the consistency of the reported results across multiple architectures and metrics suggests that the present findings remain reliable and representative within the defined dataset configuration.

The proposed framework demonstrated highly consistent performance across all evaluation metrics; however, formal statistical significance testing and multi-run variability analyses were not performed in this study. The research primarily focused on developing and validating a robust ensemble and explainable AI framework for macrofungi classification rather than conducting statistical hypothesis testing. To enhance reproducibility and quantitative validation, future investigations may include repeated experimental runs and statistical analyses, such as *t*-tests or ANOVA, to further substantiate the stability and reliability of the reported results.

The current study faced certain metadata limitations that prevented the full implementation of source-aware splitting. Although manual deduplication was carefully performed, incomplete metadata records restricted our ability to automate this process. Future datasets will benefit from improved metadata completeness and standardized acquisition protocols to enable more systematic data partitioning.

## 5. Conclusions

The findings of this study demonstrate that while several standalone deep learning models achieved strong performance in macrofungi classification, the Combination Model (EfficientNetB0 + ResNet50 + RegNetY) consistently outperformed all individual and pairwise ensembles across every evaluation metric. With an overall accuracy of 97.36%, precision and recall values exceeding 97%, an MCC of 0.9725, and an AUC of 0.9996, the Combination Model established itself as the most robust and reliable framework. These results confirm that carefully curated multi-architecture ensembles provide a powerful solution for fine-grained classification of morphologically similar mushroom species, offering both superior predictive accuracy and high interpretability through explainable AI techniques such as Grad-CAM, Eigen-CAM, and LIME.

Building on these results, future research should extend the methodology to different ensemble configurations, exploring novel combinations of CNNs and transformer-based models to further enhance performance and generalization. The framework can also be adapted for the classification of plant species, supporting applications in agriculture, forestry, and ecological monitoring. In addition, applying the approach to mushroom spore identification would address an underexplored but critical dimension of mycological taxonomy, enabling more precise differentiation at microscopic levels.

Beyond fungi and plants, the integration of this methodology with satellite imagery offers significant potential for large-scale ecological applications. By coupling ensemble-based deep learning with remote sensing data, it becomes possible to perform automated vegetation and species mapping in forest ecosystems, contributing to biodiversity assessment, conservation planning, and sustainable resource management. Such cross-disciplinary extensions not only validate the scalability of the proposed framework but also highlight its transformative potential in advancing both biological sciences and environmental monitoring.

## Figures and Tables

**Figure 1 biology-14-01644-f001:**
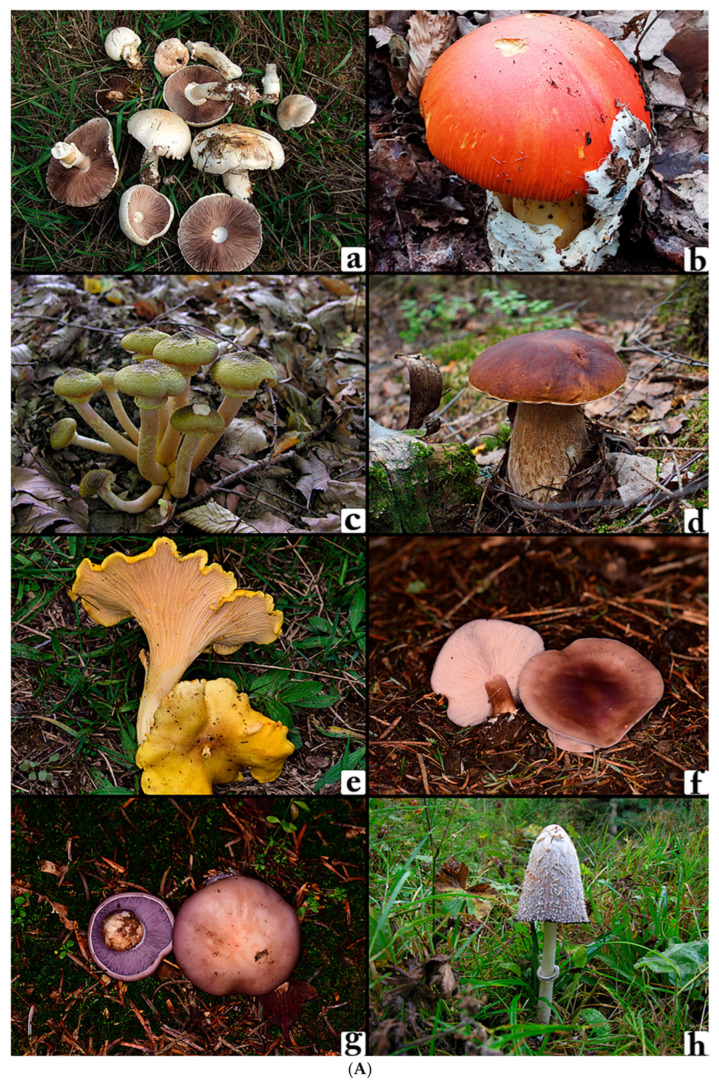
(**A**). (**a**) *Agaricus campestris*, (**b**) *Amanita caesarea*, (**c**) *Armillaria mellea*, (**d**) *Boletus edulis*, (**e**) *Cantharellus cibarius*, (**f**) *Clitocybe nebularis*, (**g**) *Collybia nuda*, (**h**) *Coprinus comatus*. (**B**). (**a**) *Craterellus cornucopioides*, (**b**) *Fistulina hepatica*, (**c**) *Hericium coralloides*, (**d**) *Hericium erinaceus*, (**e**) *Hydnum repandum*, (**f**) *Infundibulicybe geotropa*, (**g**) *Lactarius deliciosus*, (**h**) *Lactarius salmonicolor*. (**C**). (**a**) *Macrolepiota procera*, (**b**) *Marasmius oreades*, (**c**) *Morchella esculenta*, (**d**) *Pleurotus ostreatus*, (**e**) *Russula delica*, (**f**) *Sarcodon imbricatus*, (**g**) *Sparassis crispa*, (**h**) *Tricholoma terreum*.

**Figure 2 biology-14-01644-f002:**
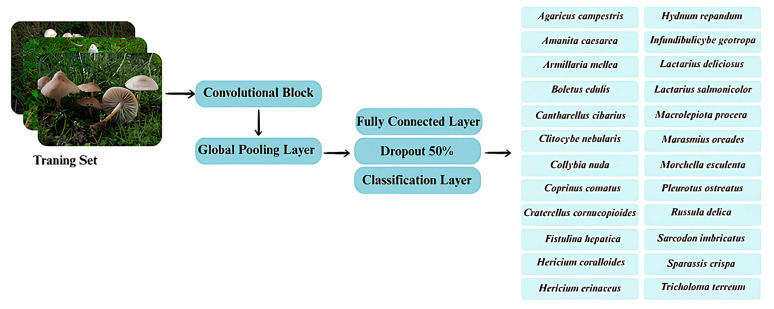
Architecture of the training pipeline classifying 24 wild edible macrofungi species using convolutional features, pooling, and dropout.

**Figure 3 biology-14-01644-f003:**
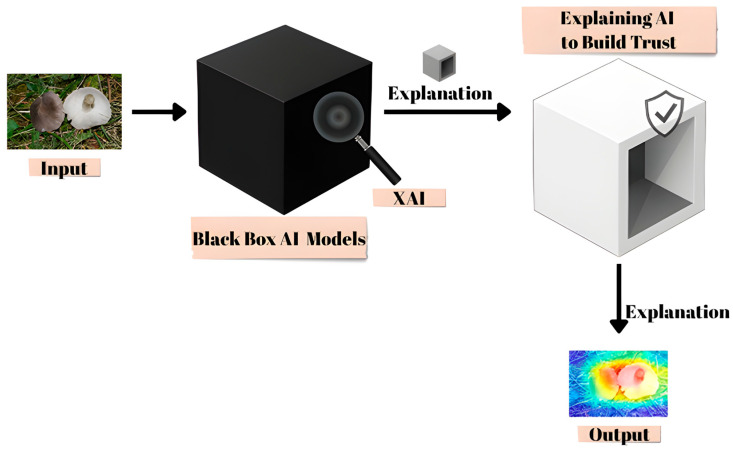
XAI reveals decision-making regions within deep learning’s black box for macrofungi classification.

**Figure 4 biology-14-01644-f004:**
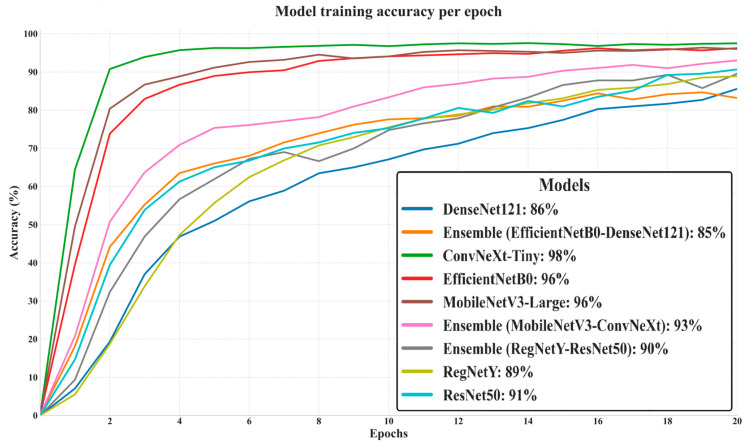
Training accuracy of deep learning models over 20 epochs.

**Figure 5 biology-14-01644-f005:**
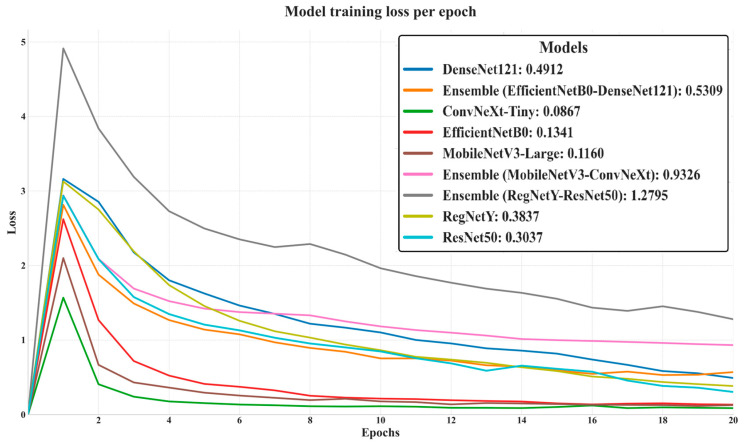
Training loss of deep learning models over 20 epochs.

**Figure 6 biology-14-01644-f006:**
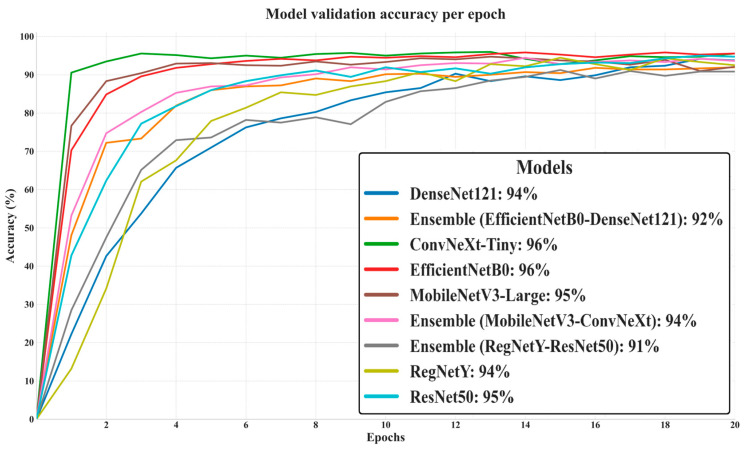
Validation accuracy of deep learning models over 20 epochs.

**Figure 7 biology-14-01644-f007:**
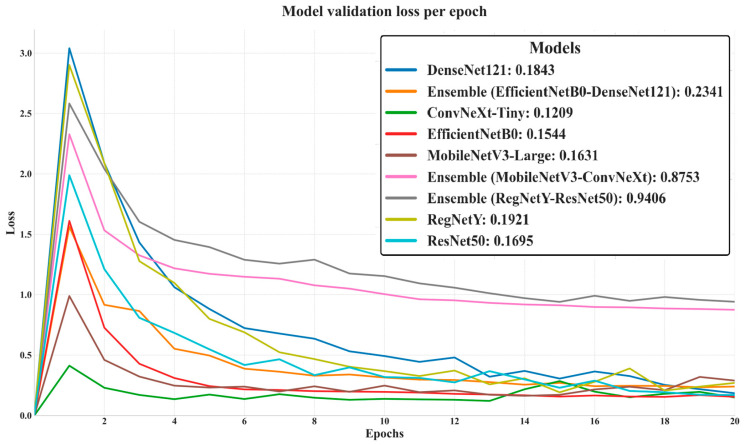
Validation loss of deep learning models over 20 epochs.

**Figure 8 biology-14-01644-f008:**
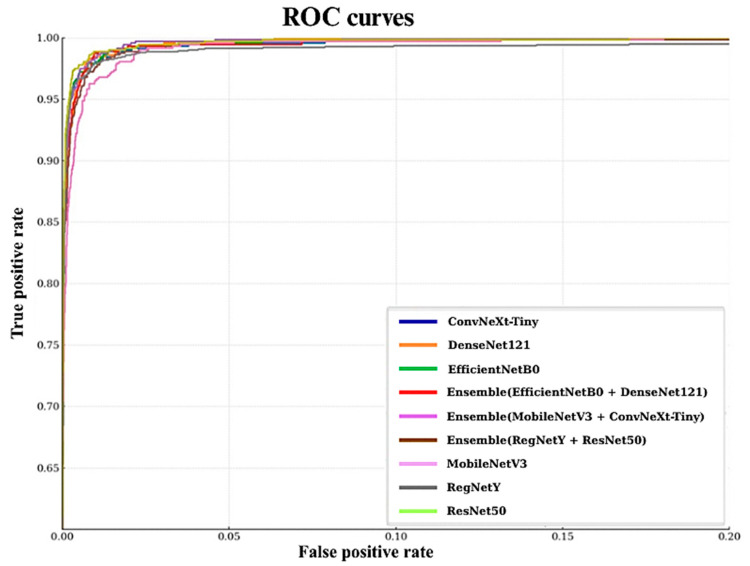
Roc curve for all models.

**Figure 9 biology-14-01644-f009:**
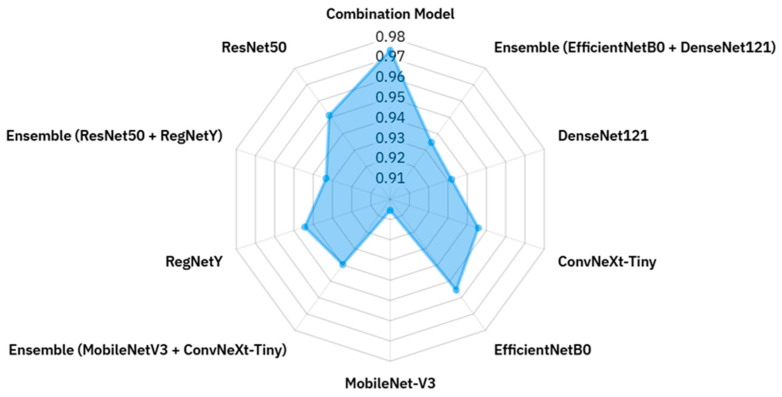
Radar chart of accuracy values for all models.

**Figure 10 biology-14-01644-f010:**
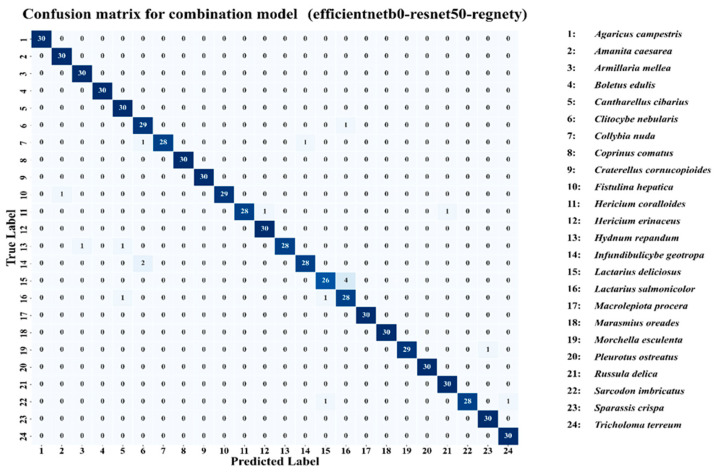
Confusion matrix for combination model (efficientnetb0-resnet50-regnety).

**Figure 11 biology-14-01644-f011:**
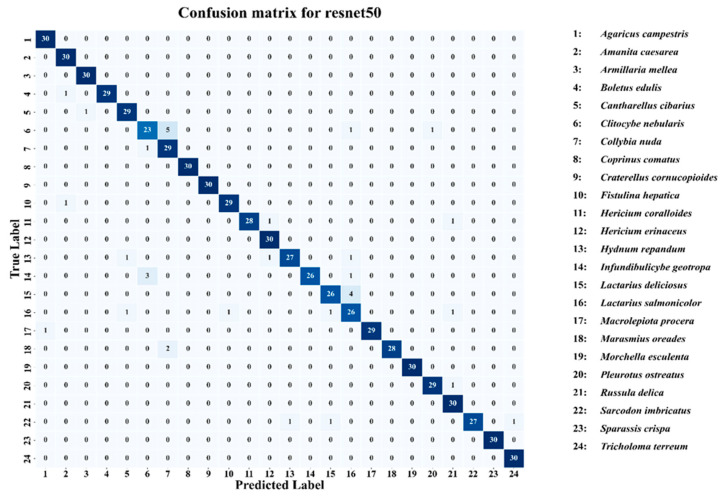
Confusion matrix for resnet50.

**Figure 12 biology-14-01644-f012:**
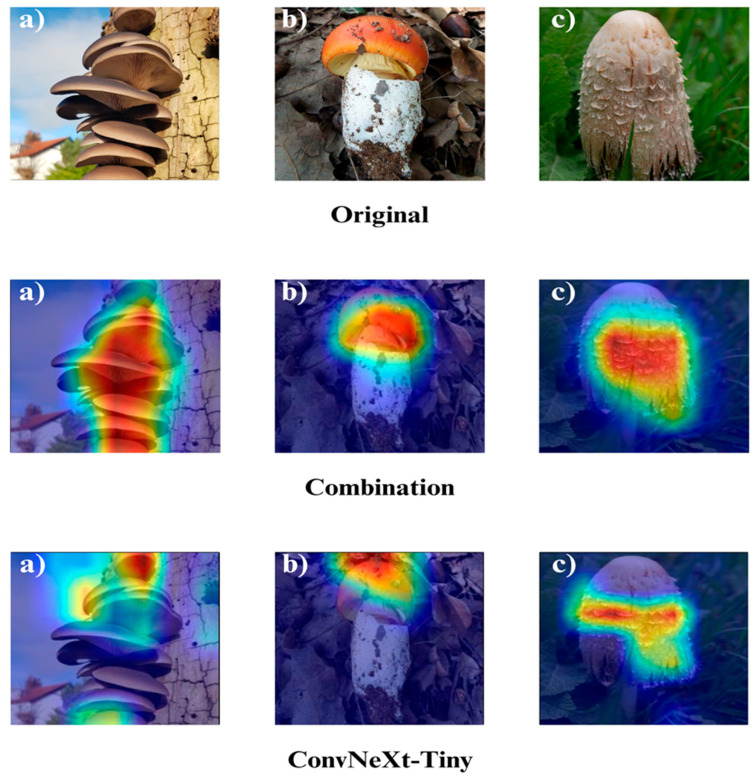
Grad-CAM visualizations for macrofungi classification using the combination model (EfficientNetB0 + ResNet50 + RegNetY) and ConvNeXt-Tiny. The images correspond to three wild edible macrofungi species: (**a**) *Pleurotus ostreatus*, (**b**) *Amanita caesarea*, and (**c**) *Coprinus comatus* from left to right. The first row displays the original images, followed by the Grad-CAM activation maps for the combination model and ConvNeXt-Tiny, respectively. These visualizations highlight the most influential regions leveraged by each model for classification, revealing how different architectures prioritize distinct structural features in the images.

**Figure 13 biology-14-01644-f013:**
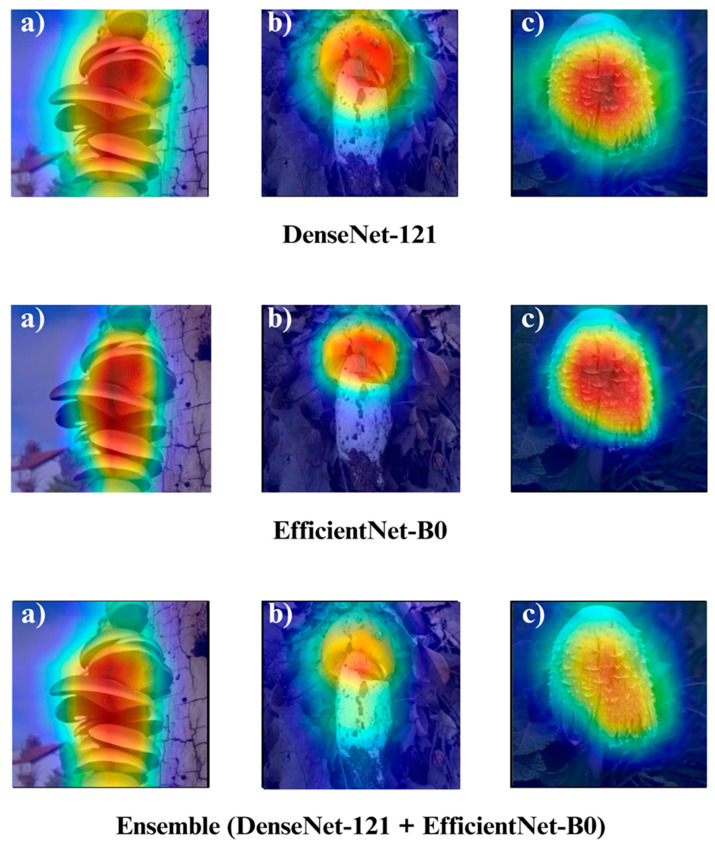
Grad-CAM visualizations for macrofungi classification using DenseNet-121, EfficientNet-B0, and the ensemble model (DenseNet-121 + EfficientNet-B0). The images represent three distinct wild edible macrofungi species: (**a**) *Pleurotus ostreatus*, (**b**) *Amanita caesarea*, and (**c**) *Coprinus comatus* from left to right. The rows display the Grad-CAM activation maps for each respective model, illustrating the specific image regions that most strongly influenced the classification decisions. These visualizations provide insight into how individual and ensemble architectures focus on different structural and textural features when identifying species.

**Figure 14 biology-14-01644-f014:**
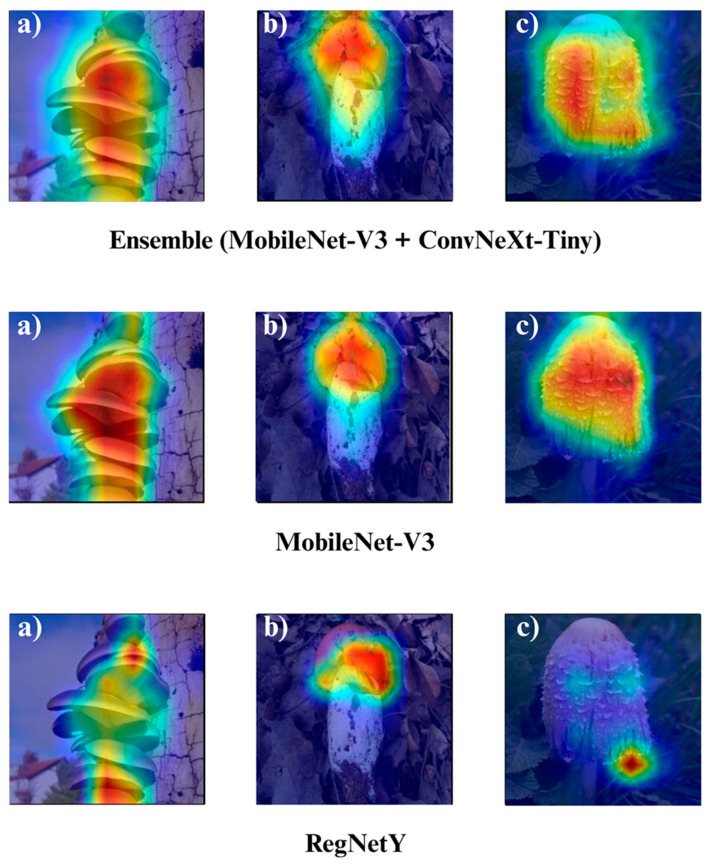
Grad-CAM visualizations for macrofungi classification using the ensemble model (MobileNet-V3 + ConvNeXt-Tiny), MobileNet-V3, and RegNetY. The images correspond to three wild edible macrofungi species: (**a**) *Pleurotus ostreatus*, (**b**) *Amanita caesarea*, and (**c**) *Coprinus comatus* from left to right. The first row shows the Grad-CAM activation maps for the ensemble model, followed by MobileNet-V3 and RegNetY. These visualizations emphasize the most critical image regions utilized by each model, illustrating the differences in how various architectures capture and interpret distinctive morphological features for classification.

**Figure 15 biology-14-01644-f015:**
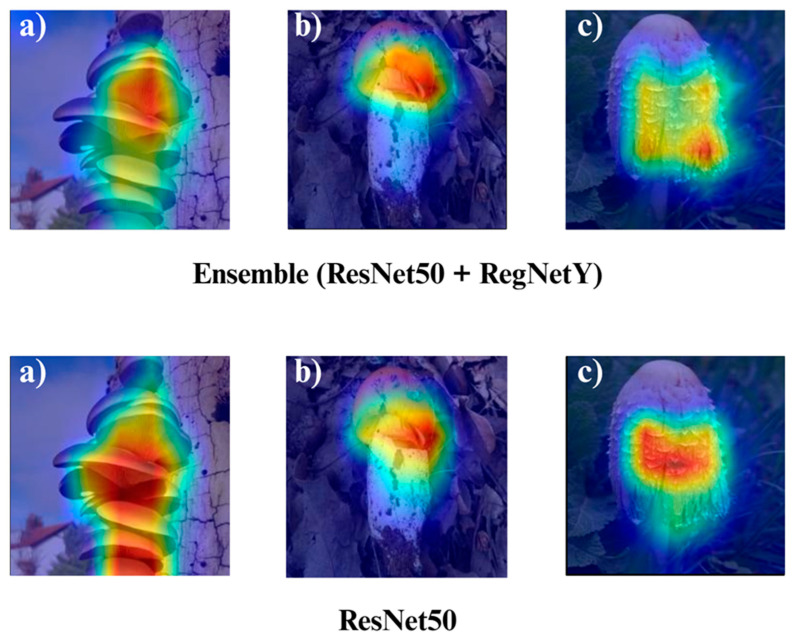
Grad-CAM visualizations for macrofungi classification using the ensemble model (ResNet50 + RegNetY) and ResNet50. The images correspond to three wild edible macrofungi species: (**a**) *Pleurotus ostreatus*, (**b**) *Amanita caesarea*, and (**c**) *Coprinus comatus* from left to right. The first row shows the Grad-CAM activation maps for the ensemble model, followed by ResNet50. These visualizations illustrate the most influential regions each model focuses on for classification, highlighting how the ensemble model and individual architectures differ in their attention to distinctive morphological features.

**Figure 16 biology-14-01644-f016:**
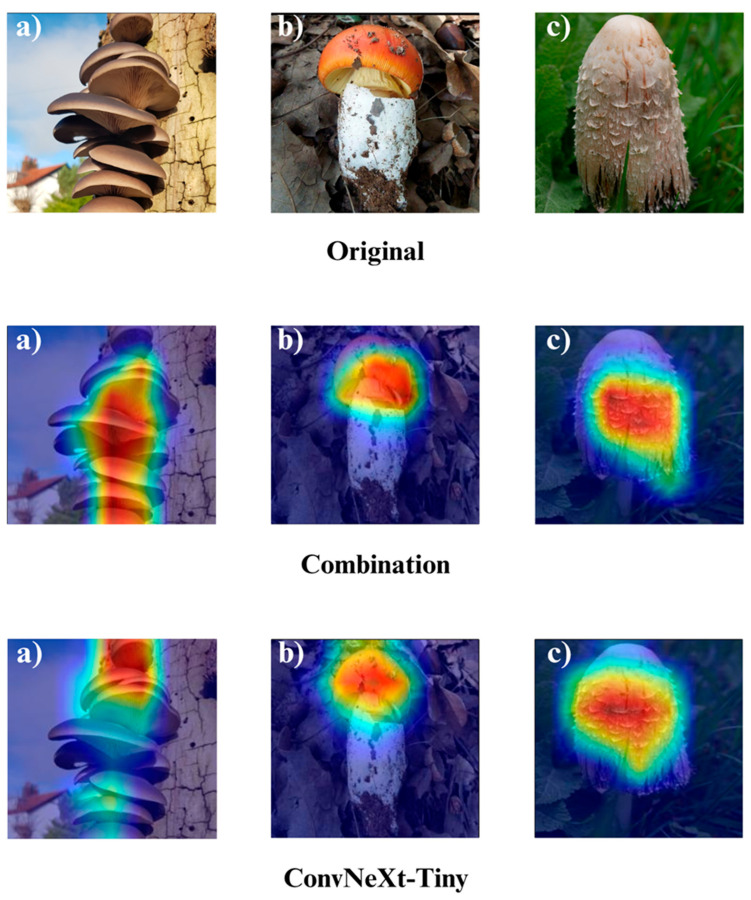
Eigen-CAM visualizations for macrofungi classification using the combination model (EfficientNetB0 + ResNet50 + RegNetY) and ConvNeXt-Tiny. The images correspond to three wild edible macrofungi species: (**a**) *Pleurotus ostreatus*, (**b**) *Amanita caesarea*, and (**c**) *Coprinus comatus* from left to right. The first row displays the original images, followed by the Eigen-CAM activation maps for the combination model and ConvNeXt-Tiny, respectively. These visualizations highlight the most influential regions leveraged by each model for classification, revealing how different architectures prioritize distinct structural features in the images.

**Figure 17 biology-14-01644-f017:**
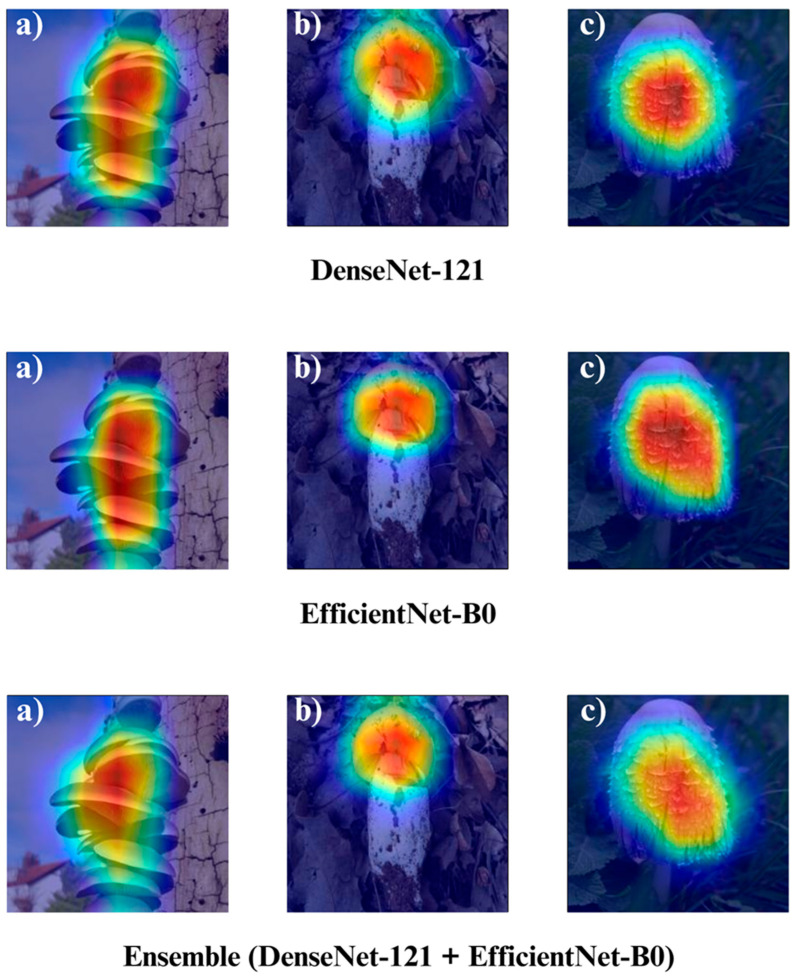
Eigen-CAM visualizations for macrofungi classification using DenseNet-121, EfficientNet-B0, and the ensemble model (DenseNet-121 + EfficientNet-B0). The images represent three distinct wild edible macrofungi species: (**a**) *Pleurotus ostreatus*, (**b**) *Amanita caesarea*, and (**c**) *Coprinus comatus* from left to right. The rows display the Eigen-CAM activation maps for each respective model, illustrating the specific image regions that most strongly influenced the classification decisions. These visualizations provide insight into how individual and ensemble architectures focus on different structural and textural features when identifying species.

**Figure 18 biology-14-01644-f018:**
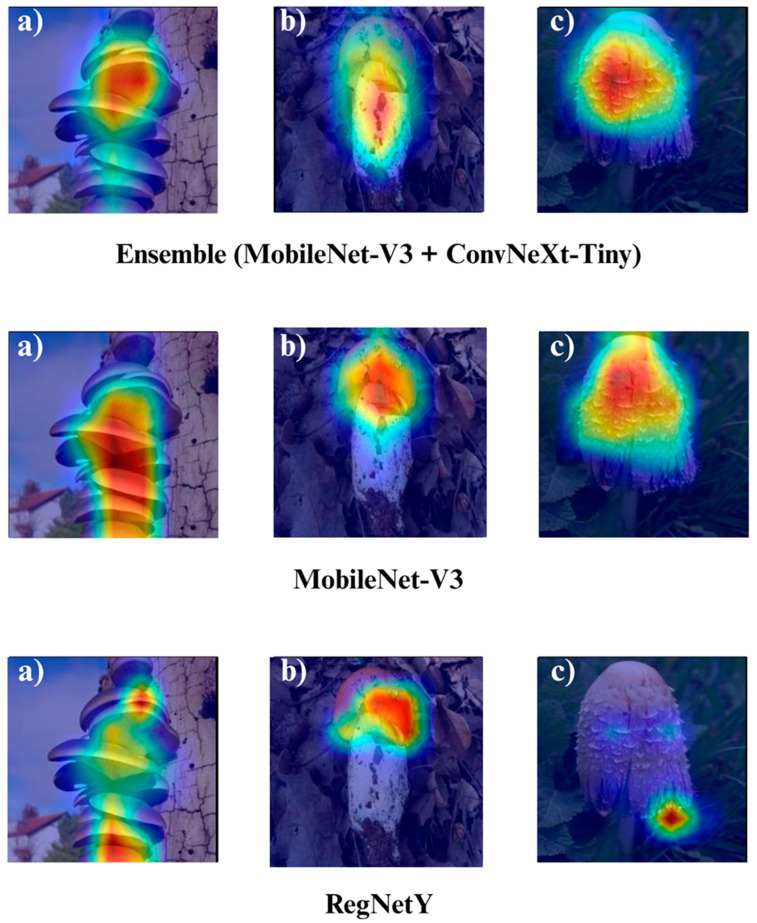
Eigen-CAM visualizations for macrofungi classification using the ensemble model (MobileNet-V3 + ConvNeXt-Tiny), MobileNet-V3, and RegNetY. The images correspond to three wild edible macrofungi species: (**a**) *Pleurotus ostreatus*, (**b**) *Amanita caesarea*, and (**c**) *Coprinus comatus* from left to right. The first row shows the Eigen-CAM activation maps for the ensemble model, followed by MobileNet-V3 and RegNetY. These visualizations emphasize the most critical image regions utilized by each model, illustrating the differences in how various architectures capture and interpret distinctive morphological features for classification.

**Figure 19 biology-14-01644-f019:**
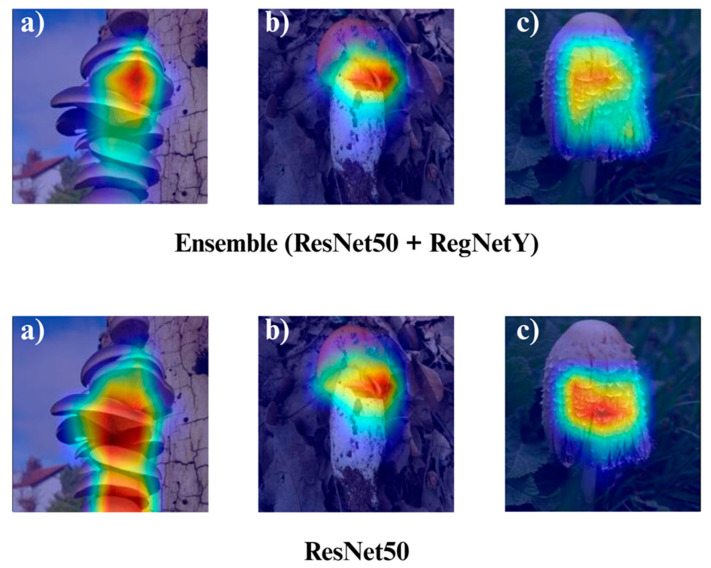
Eigen-CAM visualizations for macrofungi classification using the ensemble model (ResNet50 + RegNetY) and ResNet50. The images correspond to three wild edible macrofungi species: (**a**) *Pleurotus ostreatus*, (**b**) *Amanita caesarea*, and (**c**) *Coprinus comatus* from left to right. The first row shows the Eigen-CAM activation maps for the ensemble model, followed by ResNet50. These visualizations illustrate the most influential regions each model focuses on for classification, highlighting how the ensemble model and individual architectures differ in their attention to distinctive morphological features.

**Figure 20 biology-14-01644-f020:**
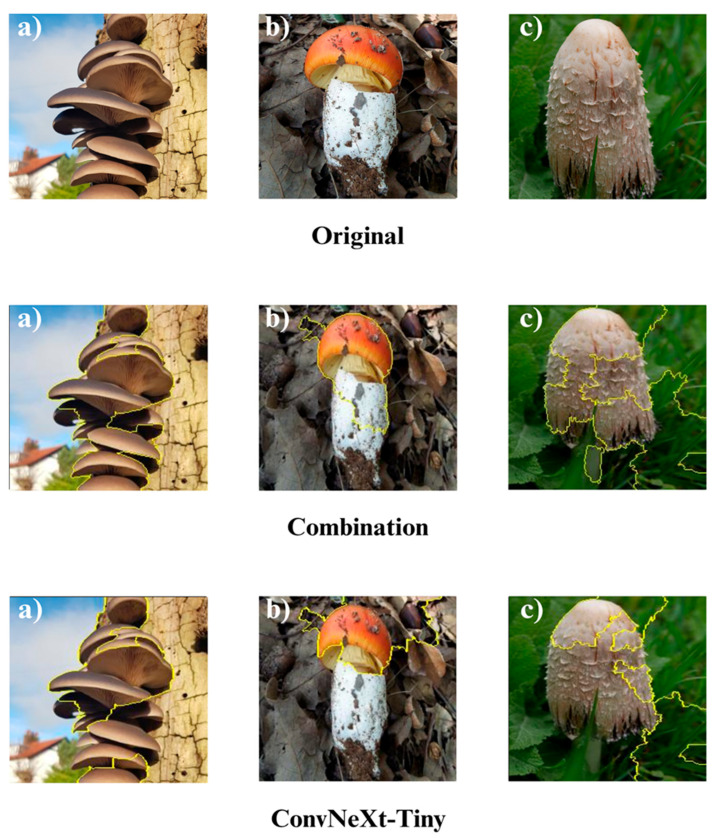
LIME (Local Interpretable Model-agnostic Explanations) visualizations for macrofungi classification using the Combination Model and ConvNeXt-Tiny architectures. The images correspond to three different wild edible macrofungi species: (**a**) *Pleurotus ostreatus*, (**b**) *Amanita caesarea*, and (**c**) *Coprinus comatus* from left to right. The yellow contours indicate the superpixel regions identified as the most influential for the classification decision, highlighting localized structural and textural cues leveraged by each model. These visualizations illustrate the models’ focus on species-specific features, with the Combination Model exhibiting more consistent and comprehensive coverage across relevant areas compared to ConvNeXt-Tiny.

**Figure 21 biology-14-01644-f021:**
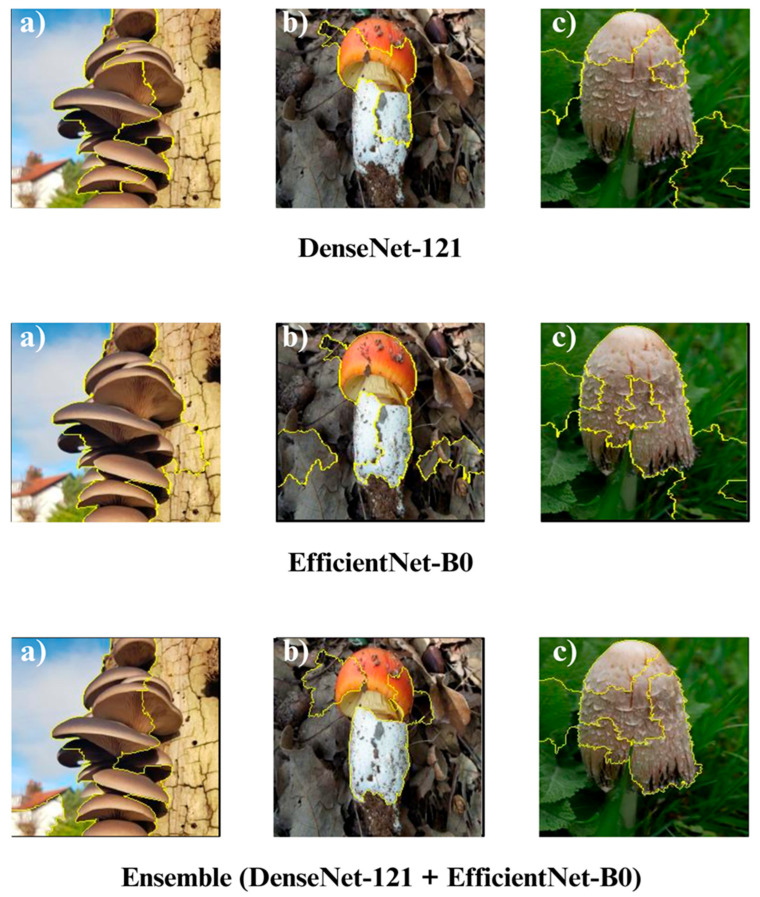
LIME (Local Interpretable Model-agnostic Explanations) visualizations for macrofungi classification using DenseNet-121, EfficientNet-B0, and their ensemble (DenseNet-121 + EfficientNet-B0). The images correspond to three different wild edible macrofungi species: (**a**) *Pleurotus ostreatus*, (**b**) *Amanita caesarea*, and (**c**) *Coprinus comatus* from left to right. The yellow contours represent the superpixel boundaries identified as most influential in the models’ classification decisions. DenseNet-121 and EfficientNet-B0 capture fine-grained morphological details relevant to each species, while the ensemble model integrates their strengths, leading to more coherent and contextually aligned focus areas across the fungal structures.

**Figure 22 biology-14-01644-f022:**
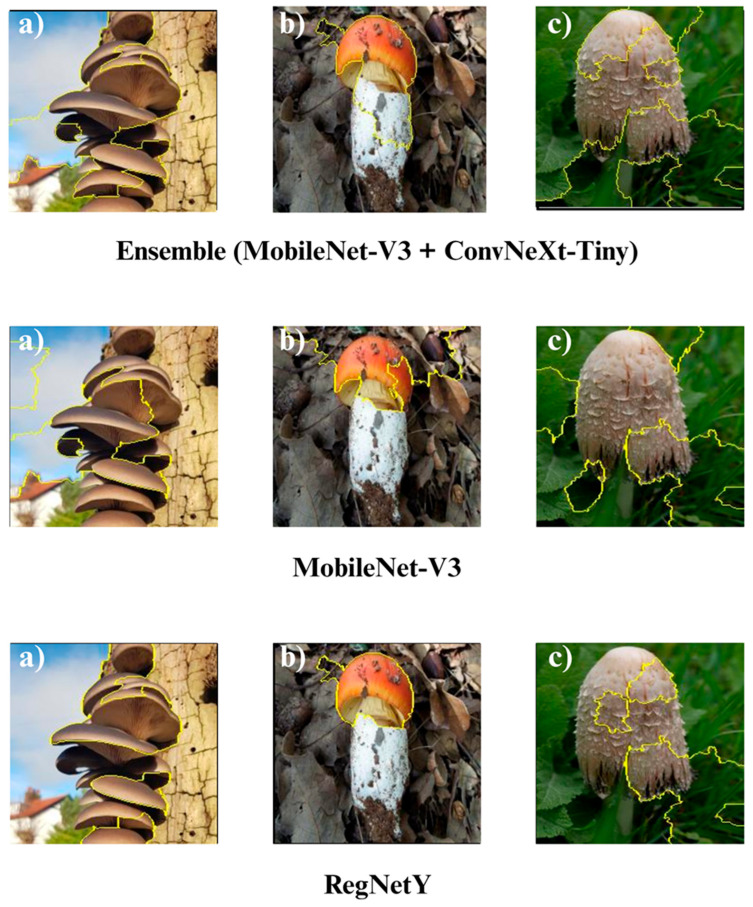
LIME visualizations for macrofungi classification using the ensemble model (MobileNet-V3 + ConvNeXt-Tiny), MobileNet-V3, and RegNetY. The images correspond to three different wild edible macrofungi species: (**a**) *Pleurotus ostreatus*, (**b**) *Amanita caesarea*, and (**c**) *Coprinus comatus*, respectively. The yellow contours delineate superpixels identified as most influential for the model’s predictions. While MobileNet-V3 and RegNetY each capture relevant structural features, such as gill arrangements, cap textures, and stem patterns, the ensemble model benefits from the complementary feature extraction capabilities of both architectures, yielding more consistent and semantically meaningful focus areas that align closely with key morphological traits of each species.

**Figure 23 biology-14-01644-f023:**
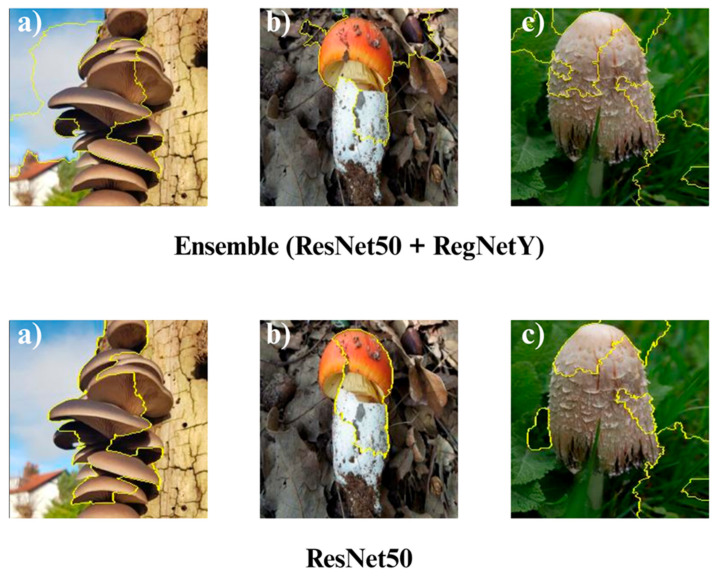
LIME visualizations for macrofungi classification using the ensemble model (ResNet50 + RegNetY) and the single ResNet50 architecture. The three columns correspond to (**a**) *Pleurotus ostreatus*, (**b**) *Amanita caesarea*, and (**c**) *Coprinus comatus*. Yellow contours mark the superpixels identified as most influential for classification. While ResNet50 effectively captures important structural cues such as gill arrangements, cap patterns, and stem textures, the ensemble approach integrates the complementary feature extraction strengths of both ResNet50 and RegNetY, leading to more stable and semantically precise localization of key morphological regions, thereby improving interpretability and potentially enhancing prediction robustness.

**Table 1 biology-14-01644-t001:** Web of Science-indexed research articles from the last five years on wild edible macrofungi classification.

Title	Reference No
First Steps in Developing a Fast, Cheap, and Reliable Method to Distinguish Wild Mushroom and Truffle Species	[9]
An Improved MobileNetV3 Mushroom Quality Classification Model Using Images with Complex Backgrounds	[10]
Detection and classification of Shiitake mushroom fruiting bodies based on Mamba YOLO	[11]
Detection of Wild Mushrooms Using Machine Learning and Computer Vision	[12]
Mushroom data creation, curation, and simulation to support classification tasks	[13]
Verified the rapid evaluation of the edible safety of wild porcini mushrooms, using deep learning and PLS-DA	[14]
Optimized DenseNet Architectures for Precise Classification of Edible and Poisonous Mushrooms	[15]
A New Deep Learning Model for the Classification of Poisonous and Edible Mushrooms Based on Improved AlexNet Convolutional Neural Network	[16]
AI-driven deep learning framework for shelf life prediction of edible mushrooms	[17]
Mushroom species classification and implementation based on improved MobileNetV3	[18]

**Table 2 biology-14-01644-t002:** Mushroom species, photograph sources, and approximate continents [37].

Mushroom Species Name	% Photos from Source	The Continents of Capture
*Agaricus campestris*	>95%	North America, Europe, Australia
*Amanita caesarea*	>95%	North America, Europe
*Armillaria mellea*	>95%	North America, Europe, Asia
*Boletus edulis*	>95%	North America, Europe, Asia, Australia
*Cantharellus cibarius*	>95%	North America, Europe, Asia
*Clitocybe nebularis*	>95%	North America, Europe, Asia
*Collybia nuda*	>95%	North America, Europe, Australia
*Coprinus comatus*	>95%	North and South America, Europe, Asia, Australia, Africa
*Craterellus cornucopioides*	>95%	North America, Europe, Australia
*Fistulina hepatica*	>95%	North America, Europe, Asia
*Hericium coralloides*	>95%	North America, Europe, Asia, Australia
*Hericium erinaceus*	>95%	North America, Europe
*Hydnum repandum*	>95%	North America, Europe, Asia, Australia
*Infundibulicybe geotropa*	>95%	Europe, Asia
*Lactarius deliciosus*	>95%	North America, Europe, Asia
*Lactarius salmonicolor*	>95%	North America, Europe
*Macrolepiota procera*	>95%	North America, Europe, Asia
*Marasmius oreades*	>95%	North America, Europe, Asia, Australia
*Morchella esculenta*	>95%	North America, Europe, Asia
*Pleurotus ostreatus*	>95%	North America, Europe, Asia
*Russula delica*	>95%	North America, Europe, Asia
*Sarcodon imbricatus*	>95%	North America, Europe
*Sparassis crispa*	>95%	North America, Europe, Asia
*Tricholoma terreum*	>95%	North America, Europe, Australia

**Table 3 biology-14-01644-t003:** Performance comparison of deep learning models based on Accuracy, Precision, Recall, Specificity, F1-Score, Mcc, G-Mean, and Auc.

Models	Accuracy	Precision	Recall	Specificity	F1-Score	Mcc	G-Mean	Auc
DenseNet121	0.9319	0.9347	0.9319	0.9970	0.9319	0.9291	0.9639	0.9987
Ensemble (EfficientNetB0-DenseNet121)	0.9347	0.9380	0.9347	0.9971	0.9350	0.9320	0.9654	0.9986
Combination (EfficientNetB0-ResNet50-RegNetY)	0.9736	0.9746	0.9736	0.9988	0.9736	0.9725	0.9861	0.9996
ConvNeXt-Tiny	0.9458	0.9517	0.9458	0.9976	0.9464	0.9437	0.9713	0.9991
EfficientNetB0	0.9555	0.9578	0.9555	0.9980	0.9551	0.9537	0.9765	0.9988
MobileNetV3-L	0.9055	0.9168	0.9055	0.9958	0.9064	0.9019	0.9496	0.9988
Ensemble (ConvNeXt-Tiny-MobileNetV3-L)	0.9444	0.9468	0.9444	0.9975	0.9439	0.9421	0.9421	0.9989
RegNetY	0.9444	0.9459	0.9444	0.9975	0.9444	0.9706	0.9706	0.9984
Ensemble (ResNet50-RegNetY)	0.9333	0.9361	0.9333	0.9971	0.9333	0.9305	0.9646	0.9978
ResNet50	0.9513	0.9539	0.9539	0.9978	0.9514	0.9527	0.9743	0.9991

## Data Availability

The complete dataset, including image identifiers, preprocessing scripts, and trained models, has been uploaded to a publicly accessible Google Drive repository and can be accessed at [https://tinyurl.com/4btbm47e] (accessed on 10 October 2025). All data are provided under the Creative Commons Attribution 4.0 License (CC BY 4.0) for academic and non-commercial use. Please see Reference [36] for the corresponding repository citation.

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
