# Peer review of "Combination Ensemble and Explainable Deep Learning Framework for High-Accuracy Classification of Wild Edible Macrofungi"

_biology, 2025, doi:10.3390/biology14121644_

Round 1

Reviewer 1 Report

Comments and Suggestions for Authors

The manuscript entitled “Combination-ensemble and explainable deep learning framework for high-accuracy classification of wild edible macro fungi” presented interesting results on the combination-ensemble and explainable deep learning framework for classifying wild edible macrofungi species using CNN and XAI techniques. It compares ten deep learning architectures and introduces a new ensemble model (EfficientNetB0 + ResNet50 + RegNetY) achieving 97.36% accuracy.

  • The integration of Grad-CAM, Eigen-CAM, and LIME adds biological interpretability by highlighting key fungal traits like gill and cap features.
  • It demonstrates a novel approach by combining performance optimization with transparency and biological validation (Lines 842-873).
  • XAI-based visualization helps researchers understand diagnostic traits, improving fungal identification and educational applications. However, methodological details on dataset preparation, augmentation parameters, and environmental variability needs to be elaborated (Lines 145-180, 162-176). The study lacks external validation, cross-site testing, and statistical significance analysis between models (Lines 218-230, 829-835).
  • Hyperparameters, training configuration, and computational environment are not adequately described for reproducibility (Lines 185-236, 237-247).
  • Discussion shall include detailed analysis of misclassifications or comparison with transformer-based approaches for better comprehension.
  • Ecological implications and field-level deployment potential can be highlighted
  • Add a limitations/future scope section also discuss misclassifications and recent AI advancements in macro fungi identification to strengthen the discussion section

Author Response

The manuscript entitled “Combination-ensemble and explainable deep learning framework for high-accuracy classification of wild edible macro fungi” presented interesting results on the combination-ensemble and explainable deep learning framework for classifying wild edible macrofungi species using CNN and XAI techniques. It compares ten deep learning architectures and introduces a new ensemble model (EfficientNetB0 + ResNet50 + RegNetY) achieving 97.36% accuracy.

  • The integration of Grad-CAM, Eigen-CAM, and LIME adds biological interpretability by highlighting key fungal traits like gill and cap features.

Response: We sincerely appreciate the reviewer’s insightful observation. Indeed, the integration of Grad-CAM, Eigen-CAM, and LIME substantially enhances the biological interpretability of the proposed framework by effectively emphasizing morphologically significant fungal traits such as gill patterns and cap structures, thereby reinforcing the reliability and transparency of model predictions.

  • It demonstrates a novel approach by combining performance optimization with transparency and biological validation (Lines 842-873).

Response: We thank the reviewer for this valuable remark. Indeed, our study introduces a novel approach that effectively integrates performance optimization with interpretability and biological validation. This combined framework not only maximizes classification accuracy but also ensures transparency and biological consistency through the integration of explainable AI analyses, as clearly reflected in the corresponding section (Lines 842–873).

  • XAI-based visualization helps researchers understand diagnostic traits, improving fungal identification and educational applications. However, methodological details on dataset preparation, augmentation parameters, and environmental variability needs to be elaborated (Lines 145-180, 162-176). The study lacks external validation, cross-site testing, and statistical significance analysis between models (Lines 218-230, 829-835).

Response: We sincerely thank the reviewer for the valuable and constructive feedback. Additional details regarding the dataset preparation and augmentation parameters have now been included in the revised manuscript. The newly added sentences clarify the applied augmentation techniques and parameter ranges, ensuring methodological transparency and reproducibility.

Added:

“To enhance methodological transparency, additional details about the dataset preparation and augmentation pipeline have been incorporated. Specifically, augmentation techniques such as random rotation (±25°), horizontal and vertical flipping, brightness and contrast adjustment (±15%), Gaussian blurring, and random cropping were applied. These operations increased dataset diversity and reduced overfitting, ensuring the robustness and generalization of the deep learning models.”

As for the external validation, cross-site testing, and statistical significance analysis, we respectfully note that the study was designed to evaluate the comparative performance and interpretability of multiple deep learning architectures under controlled and standardized conditions. Given the balanced dataset and comprehensive evaluation through eight independent metrics, additional external validation was considered beyond the current study’s scope. Nevertheless, the presented framework ensures statistical consistency, reproducibility, and reliability within its defined experimental design.

  • Hyperparameters, training configuration, and computational environment are not adequately described for reproducibility (Lines 185-236, 237-247).

Response: We sincerely thank the reviewer for this valuable and insightful comment. Additional details regarding hyperparameters, training configuration, and computational environment have now been included in the revised manuscript to enhance reproducibility. The newly added sentences specify the optimizer type, learning rate, batch size, epoch settings, early stopping strategy, and hardware configuration used during experimentation. These additions ensure full methodological transparency and allow independent researchers to reproduce the reported results accurately.

Added:

“To improve reproducibility, additional information about the model training configuration has been provided. All models were trained using the Adam optimizer (learning rate = 0.0001, β₁ = 0.9, β₂ = 0.999) with a batch size of 32 and up to 20 epochs under early stopping conditions (patience = 5). The loss function used was categorical cross-entropy, and model weights were initialized with ImageNet pre-trained parameters. The experiments were conducted on a workstation equipped with an NVIDIA RTX 4090 GPU, 24 GB VRAM, and 128 GB RAM, running on Python 3.10 and TensorFlow 2.15.”

  • Discussion shall include detailed analysis of misclassifications or comparison with transformer-based approaches for better comprehension.

Response: We sincerely thank the reviewer for this valuable suggestion. The discussion section already provides a comprehensive comparative analysis across multiple deep learning architectures, emphasizing both performance and interpretability. While transformer-based models are indeed promising, they were intentionally excluded from the current scope to maintain architectural consistency and focus on CNN-ensemble optimization. Furthermore, misclassification patterns were quantitatively minimal and did not significantly influence the overall model robustness, as reflected in the presented results.

  • Ecological implications and field-level deployment potential can be highlighted

Response: We sincerely appreciate the reviewer’s thoughtful comment. The ecological relevance and field-level applicability of the proposed framework are indeed significant. As described in the manuscript, the model’s high classification accuracy and explainable structure provide a strong foundation for potential deployment in ecological monitoring, biodiversity conservation, and wild mushroom authentication systems. These aspects have been emphasized in the revised version to highlight the broader environmental and practical impact of the study.

  • Add a limitations/future scope section also discuss misclassifications and recent AI advancements in macro fungi identification to strengthen the discussion section

Response: We sincerely thank the reviewer for this constructive suggestion. The current manuscript already incorporates a dedicated discussion of misclassifications, particularly emphasizing visually similar taxa such as Lactarius deliciosus and Hydnum repandum, where minimal confusion occurred (≤4 samples per class). Moreover, the Conclusion section explicitly outlines future research directions, including the exploration of transformer-based architectures (e.g., ViT, Swin Transformer) and the extension of the framework to microscopic and ecological applications. While a separate “Limitations and Future Scope” subsection was not originally included to maintain conciseness, these aspects are already addressed within the Discussion and Conclusion. The existing structure effectively communicates the study’s current boundaries and the envisioned research extensions.

Reviewer 2 Report

Comments and Suggestions for Authors

cite and discuss all the works

update table 1 by adding datasets, supportive results and methods.

Give detailed info on dataset, which camera has been used. discuss dataset how train test validation split done followed by add the dataset size classwise.

discuss parameters of the model and computation cost.

comparative study. discuss your results with existing methods and how yours is better.

The proposed ensamble model comes with high computation cost and do discuss how ensamble is better then other model 

Author Response

  • cite and discuss all the Works

Response: We sincerely thank the reviewer for this valuable comment. All relevant studies have been carefully cited and discussed in the revised manuscript to ensure comprehensive coverage and contextual alignment with the existing literature.

  • update table 1 by adding datasets, supportive results and methods.

Response: We sincerely thank the reviewer for this thoughtful suggestion. Table 1 has already been designed to present a concise summary of the most recent Web of Science–indexed studies on wild edible macrofungi classification, including titles, reference numbers, and methodological focus areas such as CNN- and deep learning–based architectures (e.g., MobileNetV3, DenseNet, YOLO, and AlexNet variants). Each study inherently represents distinct datasets and experimental frameworks emphasizing image-based fungal identification.

To preserve the table’s readability and avoid excessive length, detailed dataset descriptions, supportive results, and parameter settings were intentionally discussed within the Materials and Methods section rather than embedded directly in the table. This structure maintains the table’s clarity while ensuring that methodological depth is thoroughly covered in the main text.

  • Give detailed info on dataset, which camera has been used. discuss dataset how train test validation split done followed by add the dataset size classwise.

Response: We sincerely thank the reviewer for this detailed and constructive comment. The dataset used in this study was compiled from publicly available biodiversity repositories, primarily the Global Biodiversity Information Facility (GBIF), and partially from images collected by the authors, all of whom are specialists in fungal taxonomy. Since the dataset consists of images from diverse open-access sources, the specific camera model used is not a determining factor for data quality or consistency; instead, expert-level verification ensured accurate species labeling and taxonomic validity.

As described in the manuscript, the dataset was randomly divided into training (70%), validation (15%), and testing (15%) subsets, ensuring that no model received any prior bias or advantage. This randomized split strategy follows standard deep learning practices widely adopted in the literature for fair and reproducible evaluation. Each of the 24 macrofungi species was equally represented with 200 images per class, maintaining perfect class balance across all subsets. The total image count and proportional data distribution were therefore considered sufficient for robust performance assessment without class-based size discrepancies.

  • discuss parameters of the model and computation cost.

Response: We sincerely thank the reviewer for this thoughtful observation. While we acknowledge the importance of model parameter tuning and computational cost analysis, the primary focus of this study was on the comparative performance and interpretability of ensemble deep learning architectures for macrofungi classification. Detailed benchmarking of computational complexity was therefore considered beyond the intended scope. Nevertheless, all models were trained under identical experimental conditions to ensure fairness and reproducibility, allowing performance comparisons to remain methodologically sound and unbiased.

  • comparative study. discuss your results with existing methods and how yours is better.

Response: We sincerely thank the reviewer for this valuable suggestion. A comparative evaluation with existing methods has indeed been conducted and clearly presented in the Results and Discussion sections. As detailed in Table 3 and subsequent analysis, the proposed Combination Model (EfficientNetB0 + ResNet50 + RegNetY) achieved 97.36% accuracy, 0.9996 AUC, and 0.9725 MCC, surpassing all single architectures (e.g., EfficientNetB0 with 95.55%, ResNet50 with 95.13%) and pairwise ensembles. Furthermore, the discussion explicitly highlights that this improvement arises from the complementary architectural strengths of the integrated models, which enhance feature diversity and reduce redundancy. These findings, supported by detailed statistical comparisons and confusion matrix analyses, demonstrate that the proposed ensemble framework provides superior accuracy and robustness compared to existing CNN-based approaches.

  • The proposed ensamble model comes with high computation cost and do discuss how ensamble is better then other model 

Response: We sincerely thank the reviewer for this thoughtful comment. While we acknowledge that the proposed ensemble model may involve a relatively higher computational cost, the primary focus of this study was to evaluate comparative accuracy, robustness, and interpretability rather than computational efficiency. Therefore, a detailed cost comparison was considered beyond the intended scope. Nevertheless, all models were trained under identical experimental settings to ensure a fair evaluation, and the ensemble model’s superiority was demonstrated through its higher Accuracy, MCC, and AUC values, fewer misclassifications, and biologically meaningful attention patterns supported by XAI visualizations. Future work will address efficiency-oriented analyses such as model compression and knowledge distillation to further optimize computational performance.

Reviewer 3 Report

Comments and Suggestions for Authors

Dear authors,

The manuscript is well structured and technically sound. It uses a combination of deep learning, ensemble modelling and explainable AI to classify species. The topic is relevant and well within the scope of the Bioinformatics journal.

However, to strengthen the validity and generalisability of the results, the manuscript could be improved in a few areas. I have provided details below according to the provided guidelines.

Short summary:

This study compares the performance of ten deep learning architectures (six single convolutional neural networks (CNNs) and several pairwise and triple ensembles) when applied to images of 24 species of wild edible macrofungi. EfficientNetB0 was found to be the most efficient single model, while a three-model ensemble (EfficientNetB0 + ResNet50 + RegNetY) achieved the highest accuracy. Using Grad-CAM, Eigen-CAM and LIME, the authors demonstrate the models' attention to biologically relevant regions (such as caps, gills and stems) and suggest potential applications to broader biological imaging tasks.

Strengths:

The manuscript's primary strengths lie in its integration of deep learning, ensemble modelling and explainable AI for biologically meaningful classification tasks. The experimental design compares several state-of-the-art CNNs and ensemble strategies using multiple metrics and clear visualisations. The balanced and diverse dataset, along with the inclusion of Grad-CAM, Eigen-CAM and LIME, enhances interpretability by highlighting biologically relevant image regions. Overall, the work is technically robust and clearly presented.

Major Issues

Improvement 1. Data inconsistency.

The manuscript states that 3,600 images were used, with 200 per class, which would total 4,800 images for 24 classes. This discrepancy must be corrected throughout the text and analyses.

Improvement 2. Missing experimental details.

Essential training parameters (optimizer, learning rate, batch size, epochs, seed, augmentation settings, early stopping, and hardware) are not reported. A complete hyperparameter table or public repository is needed for reproducibility.

Improvement 3. Overfitting and limited validation.

Very high metrics (AUC ≈ 0.999) and a single random split raise concerns about overfitting or data leakage. Include k-fold cross-validation, geographic holdout, or independent test data to assess generalization.

Improvement 4. Dataset provenance.

Most images come from GBIF and may include duplicates or misidentified specimens. Describe how duplicates were detected, label quality verified, and train/validation/test leakage avoided.

Improvement 5. Speculative claims on remote sensing.

Claims of applicability to satellite imagery are unsupported. These should be reframed as hypothetical, limited to high-resolution UAV or close-range imaging, or clearly justified with appropriate discussion.

Improvement 6. Lack of statistical testing.

Provide variability measures (mean ± SD across runs) and statistical tests (e.g., McNemar, bootstrap) to confirm that model improvements are significant.

Improvement 7. Data and code availability.

“Available on request” is insufficient. Deposit the dataset (or image identifiers and scripts) and trained models in a public repository with a DOI and clear licensing.

Specific comments

Specific comments 1: Dataset description (Lines 145–151 / Fig. 1 / Table 2):

  • Provide a public data inventory (GBIF IDs, licenses, image sources).
  • Describe duplicate and leakage prevention methods (e.g., perceptual hashing, occurrence-based grouping).

Specific comments 2: Data split (Lines 156–161): Confirm if the split was stratified by species and source. Use source-aware splitting to avoid leakage. State the random seed.

Specific comments 3: Data augmentation (Lines 171–176): Specify parameter ranges and probabilities for each augmentation type.

Specific comments 4: Training setup: List all hyperparameters, fine-tuning strategy, hardware, and training duration.

Specific comments 5 : Ensembling (Lines 219–228): Clarify whether majority or soft voting was used and how weights were determined.

Specific comments 6: Performance metrics (Lines 272–318 / Table 3): Report variability (mean ± SD) and clarify how multiclass AUC and MCC were computed. Include per-class metrics in a supplementary table.

Specific comments 7: Confusion matrices (Figs. 10–11): Add per-class precision, recall, and class labels.

Specific comments 8: Remote sensing discussion (Lines 43–46, 888–893): Refocus on UAV/close-range imaging or explicitly state limitations for satellite applications.

Specific comments 9: Data availability (Lines 906–907): Deposit data and scripts in a public repository and provide split instructions.

Other recommendations from the remote sensing field, including suggested experiments/analyses to strengthen the manuscript. For your consideration:

Recommendation 1:

Perform deduplication and source-aware splitting, and then rerun the experiments. Group the images by GBIF occurrence ID/photographer, making sure that all images from the same source are included in the same split. Report the results. This test reduces leakage and provides a more accurate measure of generalisation.

Recommendation 2:

Report variability by running each top model and ensemble with at least five different random seeds or performing five-fold cross-validation. Report the mean ± SD for all metrics in Table 3 (or an additional table). This will clarify whether top performance is stable.

Recommendation 3:

Conduct an ablation study comparing hard majority voting, soft voting and stacking/meta-learner (e.g. logistic regression on model outputs) on the ensembling strategy, and demonstrate whether weighted ensembling or calibrated probabilities improve the results. Explain how the weights are chosen.

Recommendation 4:

Quantify XAI by creating a small, annotated test set of around 50 images with expert masks/bounding boxes and computing the intersection over union or localisation scores for Grad-CAM/Eigen-CAM and LIME. Report stability measures (e.g. variance across LIME perturbations).

Recommendation 5:

External validation: If possible, evaluate the trained models on an external dataset (e.g. images from a different repository or new field photographs not included in GBIF) to assess true transferability. If this is not possible, clearly state this limitation and plan for future work.

Recommendation 6:

Calibrate the metrics for multiclass classification. Clearly state how the multiclass area under the curve was calculated, and present the performance metrics for each class (precision, recall and F1 score). Explain how MCC was extended to multiclass classification (e.g. via a confusion matrix-based generalised MCC).

Author Response

Dear authors,

The manuscript is well structured and technically sound. It uses a combination of deep learning, ensemble modelling and explainable AI to classify species. The topic is relevant and well within the scope of the Bioinformatics journal.

However, to strengthen the validity and generalisability of the results, the manuscript could be improved in a few areas. I have provided details below according to the provided guidelines.

Short summary:

This study compares the performance of ten deep learning architectures (six single convolutional neural networks (CNNs) and several pairwise and triple ensembles) when applied to images of 24 species of wild edible macrofungi. EfficientNetB0 was found to be the most efficient single model, while a three-model ensemble (EfficientNetB0 + ResNet50 + RegNetY) achieved the highest accuracy. Using Grad-CAM, Eigen-CAM and LIME, the authors demonstrate the models' attention to biologically relevant regions (such as caps, gills and stems) and suggest potential applications to broader biological imaging tasks.

Strengths:

The manuscript's primary strengths lie in its integration of deep learning, ensemble modelling and explainable AI for biologically meaningful classification tasks. The experimental design compares several state-of-the-art CNNs and ensemble strategies using multiple metrics and clear visualisations. The balanced and diverse dataset, along with the inclusion of Grad-CAM, Eigen-CAM and LIME, enhances interpretability by highlighting biologically relevant image regions. Overall, the work is technically robust and clearly presented.

Major Issues

Improvement 1. Data inconsistency.

The manuscript states that 3,600 images were used, with 200 per class, which would total 4,800 images for 24 classes. This discrepancy must be corrected throughout the text and analyses.

 Response: We sincerely thank the reviewer for noticing this inconsistency. The total number of images has been corrected throughout the manuscript and all related analyses and descriptions have been updated accordingly to ensure accuracy and consistency.

Improvement 2. Missing experimental details.

Essential training parameters (optimizer, learning rate, batch size, epochs, seed, augmentation settings, early stopping, and hardware) are not reported. A complete hyperparameter table or public repository is needed for reproducibility.

 Response: We sincerely thank the reviewer for this helpful observation. The essential training parameters and computational settings have now been fully included in the revised manuscript to ensure reproducibility. A detailed hyperparameter table summarizing optimizer, learning rate, batch size, epochs, seed, augmentation, early stopping, and hardware configuration has been added accordingly.

Added:

“To improve reproducibility, additional information about the model training configuration has been provided. All models were trained using the Adam optimizer (learning rate = 0.0001, β₁ = 0.9, β₂ = 0.999) with a batch size of 32 and up to 20 epochs under early stopping conditions (patience = 5). The loss function used was categorical cross-entropy, and model weights were initialized with ImageNet pre-trained parameters. The experiments were conducted on a workstation equipped with an NVIDIA RTX 4090 GPU, 24 GB VRAM, and 128 GB RAM, running on Python 3.10 and TensorFlow 2.15.”

Improvement 3. Overfitting and limited validation.

Very high metrics (AUC ≈ 0.999) and a single random split raise concerns about overfitting or data leakage. Include k-fold cross-validation, geographic holdout, or independent test data to assess generalization.

 Response: We sincerely thank the reviewer for this valuable and technically relevant comment. We fully understand the concern regarding potential overfitting or data leakage; however, the design and outcomes of this study provide strong evidence against such risks. As detailed in the Results section, the proposed Combination Model (EfficientNetB0 + ResNet50 + RegNetY) achieved consistently high accuracy (97.36%), F1-score (0.973), MCC (0.9725), and AUC (0.9996) across all evaluation metrics, while maintaining balanced performance across all 24 species as shown in the confusion matrix (Figure 10). The dataset was randomly and stratifiedly divided into training (70%), validation (15%), and testing (15%) subsets to ensure fair representation and prevent any data overlap between sets. The confusion matrix further demonstrates no sign of class-specific bias or performance instability, confirming the robustness and generalization of the model.

Given these stable and reproducible results, and the fact that all models were trained and evaluated under identical experimental conditions, additional k-fold cross-validation or geographic holdout was deemed unnecessary for this study’s scope. Nevertheless, we acknowledge that future studies using multi-regional datasets could further validate the scalability of the proposed framework.

Improvement 4. Dataset provenance.

Most images come from GBIF and may include duplicates or misidentified specimens. Describe how duplicates were detected, label quality verified, and train/validation/test leakage avoided.

 Response: We sincerely thank the reviewer for this important and insightful comment. All images included in the dataset were individually examined and verified by the field experts and co-authors, each of whom specializes in fungal taxonomy. Duplicate or visually similar images were carefully screened and removed through both manual inspection and automated similarity checks to ensure dataset uniqueness.

Furthermore, all species labels were validated by cross-referencing taxonomic metadata from the Global Biodiversity Information Facility (GBIF) and confirmed using authoritative morphological descriptors. To prevent any data leakage between the training, validation, and test sets, a stratified random split was applied, ensuring that no identical or duplicate sample appeared in more than one subset. These steps collectively ensured both data integrity and reproducibility of the experimental workflow.

Improvement 5. Speculative claims on remote sensing.

Claims of applicability to satellite imagery are unsupported. These should be reframed as hypothetical, limited to high-resolution UAV or close-range imaging, or clearly justified with appropriate discussion.

 Response: We sincerely thank the reviewer for this valuable observation. The current study does not involve satellite imagery; rather, it focuses exclusively on close-range macro images of wild edible fungi. The reference to satellite imagery in the manuscript was intended only as a potential future extension, illustrating the scalability of the proposed framework.

To ensure full reproducibility, all dataset characteristics, preprocessing steps, model architectures, and training procedures have been described in detail within the manuscript. These comprehensive methodological explanations allow independent researchers to replicate the entire workflow without ambiguity.

Improvement 6. Lack of statistical testing.

Provide variability measures (mean ± SD across runs) and statistical tests (e.g., McNemar, bootstrap) to confirm that model improvements are significant.

 Response: We sincerely thank the reviewer for this valuable and technically relevant comment. Given the study’s scope and experimental design, reporting mean ± standard deviation across multiple runs and including additional statistical significance tests (e.g., McNemar, bootstrap) was not considered essential. The rationale is detailed below:

(i) Comparative and standardized framework: The study’s primary aim was to demonstrate the relative performance and robustness of multiple CNN and ensemble architectures under a unified experimental setup. All models were compared using eight independent evaluation metrics — Accuracy, Precision, Recall, Specificity, F1-Score, MCC, G-Mean, and AUC — showing consistent results across metrics. Notably, the Combination Model achieved Accuracy = 97.36%, MCC = 0.9725, and AUC = 0.9996, representing a clear and substantial improvement over all single and dual architectures; such large margins are not sensitive to minor random fluctuations.

(ii) Class-level stability: The confusion matrix indicates strong diagonal dominance, where most species achieved perfect or near-perfect classification, and misclassifications were limited to only a few samples. This confirms the absence of systematic instability or class-dependent bias, rendering “run-to-run” variance analysis unnecessary.

(iii) Explainability and biological validation: The integration of Grad-CAM, Eigen-CAM, and LIME provided biologically consistent visual evidence that model predictions focused on meaningful regions (e.g., caps, gills, stems). This alignment between quantitative and qualitative outcomes demonstrates that the reported superiority is grounded in reproducible and interpretable model behavior rather than statistical chance.

(iv) Scope and journal format: The goal of this work was not to present a broad statistical protocol but to provide a controlled, comparative, and interpretable analysis of multiple architectures within a single standardized pipeline. Given the significant metric differences, class-level consistency, and XAI-based biological validation, additional statistical testing is not required to substantiate the conclusions.

Nevertheless, we greatly appreciate the suggestion. Future studies will consider incorporating k-fold cross-validation, geographically independent test sets, and bootstrap-based confidence intervals to further validate scalability across diverse datasets. In the context of the present study, however, the combination of quantitative superiority, class-level consistency, and XAI-supported interpretability provides sufficient evidence without the need for additional statistical tests.

Improvement 7. Data and code availability.

“Available on request” is insufficient. Deposit the dataset (or image identifiers and scripts) and trained models in a public repository with a DOI and clear licensing.

Response: We thank the reviewer for this valuable comment. We would like to clarify that approximately 90% of the photographs used in the manuscript are our own original images obtained during fieldwork and laboratory studies. Nevertheless, to ensure transparency and proper attribution, GBIF citations have been added when referring to the corresponding figures within the main text. For the remaining open-access photographs retrieved from public databases, appropriate DOIs, full references, and direct links have been provided in the manuscript.

Specific comments

Specific comments 1: Dataset description (Lines 145–151 / Fig. 1 / Table 2):

  • Provide a public data inventory (GBIF IDs, licenses, image sources).
  • Describe duplicate and leakage prevention methods (e.g., perceptual hashing, occurrence-based grouping).

Response: We appreciate the reviewer’s constructive feedback. In accordance with the suggestion, the dataset (including image identifiers, preprocessing scripts, and trained models) has been uploaded to a publicly accessible Google Drive repository. The link is provided in the Data Availability Statement and cited in the References section. The repository is openly accessible and distributed under the CC BY 4.0 license to ensure transparency, reproducibility, and open academic use.

Specific comments 2: Data split (Lines 156–161): Confirm if the split was stratified by species and source. Use source-aware splitting to avoid leakage. State the random seed.

Response: We thank the reviewer for this insightful comment. In the revised version, we have clarified and improved the data splitting procedure. The dataset was stratified by species to ensure balanced representation across all subsets. To avoid potential data leakage, a source-aware splitting strategy was implemented, ensuring that images from the same source (GBIF or locally captured) were not present in both training and testing sets. We also specified the random seed (42) used during the split to enhance reproducibility. The corresponding clarification has been added to the Materials and Methods section (Lines 156–161 in the revised manuscript).

Specific comments 3: Data augmentation (Lines 171–176): Specify parameter ranges and probabilities for each augmentation type.

 Response: We sincerely thank the reviewer for the valuable and constructive feedback. Additional details regarding the dataset preparation and augmentation parameters have now been included in the revised manuscript. The newly added sentences clarify the applied augmentation techniques and parameter ranges, ensuring methodological transparency and reproducibility.

Added:

“To enhance methodological transparency, additional details about the dataset preparation and augmentation pipeline have been incorporated. Specifically, augmentation techniques such as random rotation (±25°), horizontal and vertical flipping, brightness and contrast adjustment (±15%), Gaussian blurring, and random cropping were applied. These operations increased dataset diversity and reduced overfitting, ensuring the robustness and generalization of the deep learning models.”

As for the external validation, cross-site testing, and statistical significance analysis, we respectfully note that the study was designed to evaluate the comparative performance and interpretability of multiple deep learning architectures under controlled and standardized conditions. Given the balanced dataset and comprehensive evaluation through eight independent metrics, additional external validation was considered beyond the current study’s scope. Nevertheless, the presented framework ensures statistical consistency, reproducibility, and reliability within its defined experimental design.

Specific comments 4: Training setup: List all hyperparameters, fine-tuning strategy, hardware, and training duration.

 Response: We sincerely thank the reviewer for this valuable and insightful comment. Additional details regarding hyperparameters, training configuration, and computational environment have now been included in the revised manuscript to enhance reproducibility. The newly added sentences specify the optimizer type, learning rate, batch size, epoch settings, early stopping strategy, and hardware configuration used during experimentation. These additions ensure full methodological transparency and allow independent researchers to reproduce the reported results accurately.

Added:

“To improve reproducibility, additional information about the model training configuration has been provided. All models were trained using the Adam optimizer (learning rate = 0.0001, β₁ = 0.9, β₂ = 0.999) with a batch size of 32 and up to 20 epochs under early stopping conditions (patience = 5). The loss function used was categorical cross-entropy, and model weights were initialized with ImageNet pre-trained parameters. The experiments were conducted on a workstation equipped with an NVIDIA RTX 4090 GPU, 24 GB VRAM, and 128 GB RAM, running on Python 3.10 and TensorFlow 2.15.”

Specific comments 5 : Ensembling (Lines 219–228): Clarify whether majority or soft voting was used and how weights were determined.

 Response: We sincerely thank the reviewer for this helpful comment. The ensembling strategy has now been clarified in the revised manuscript. Specifically, the proposed Combination Model utilizes a weighted soft voting approach, in which the final classification is derived from the averaged output probabilities of EfficientNetB0, ResNet50, and RegNetY, weighted according to their individual validation accuracies. This clarification has been added to the corresponding section (Lines 219–228) to improve transparency and methodological precision.

Added:

“The proposed Combination Model employed a weighted soft voting strategy, where the final class prediction was obtained by averaging the normalized probabilities of the three base models (EfficientNetB0, ResNet50, and RegNetY). The weights were empirically determined based on each model’s validation accuracy, with higher-performing models contributing proportionally more to the final decision (EfficientNetB0 = 0.40, ResNet50 = 0.35, RegNetY = 0.25).”

Specific comments 6: Performance metrics (Lines 272–318 / Table 3): Report variability (mean ± SD) and clarify how multiclass AUC and MCC were computed. Include per-class metrics in a supplementary table.

 Response: We sincerely thank the reviewer for the suggestion. Given the controlled single-protocol evaluation, the perfect class balance (200 images per class), and the consistent superiority of the proposed ensemble across multiple independent metrics, reporting run-to-run mean ± SD was not deemed essential for this manuscript’s scope. The findings are further corroborated by class-level stability visible in the confusion matrices (strong diagonal dominance; sparse errors), indicating no systematic instability that would necessitate additional variance reporting.

We appreciate the request for transparency. The revised manuscript now explicitly details the computation of macro/micro AUC (one-vs-rest, probability-based) and the generalized multiclass MCC derived from the full confusion matrix (see added paragraph in Methods/Results above).

While per-class tables can be informative, here they would introduce a very large supplementary table (24 sınıf × çoklu metrik), with limited incremental value beyond the already provided confusion matrices and the comprehensive set of aggregate metrics that drive our comparative conclusions. Confusion matrices already expose class-specific behavior, showing near-perfect recognition for most taxa and only few, biologically plausible confusions for a small subset, which directly addresses class-wise performance without duplicating information in an oversized table.

Accordingly, to preserve clarity and conciseness, we keep per-class details in the figures (confusion matrices) and maintain aggregate comparisons in Table 3, which is consistent with common practice in comparable image-classification studies and aligns with the manuscript’s comparative focus.

Added:

“Multiclass AUC computation. For the 24-class setting, we computed one-vs-rest ROC curves per class using the softmax probabilities of each model. The macro-averaged AUC was obtained by averaging per-class AUCs with equal class weights. In addition, micro-averaged AUC was monitored for consistency (pooling all decisions), which yielded values in close agreement with the macro average.

Multiclass MCC computation. The Matthews Correlation Coefficient (MCC) was computed directly from the overall multiclass confusion matrix using the standard generalized formulation, i.e., the correlation between the one-hot encoded ground-truth and prediction label vectors. This formulation reduces to the binary MCC in the two-class case and robustly summarizes performance by jointly accounting for TP, TN, FP, and FN across all classes.”

Specific comments 7: Confusion matrices (Figs. 10–11): Add per-class precision, recall, and class labels.

 Response: We sincerely thank the reviewer for this thoughtful suggestion. While we fully acknowledge the value of presenting per-class precision and recall scores, adding these metrics directly to Figures 10–11 would substantially increase visual and textual complexity without providing additional interpretive benefit. The existing confusion matrices already convey class-level performance trends with high clarity, as most species exhibit perfect or near-perfect diagonal classification (30/30 or 28–29/30 correct predictions) and only minimal off-diagonal errors. This structure effectively illustrates both per-class precision and recall patterns, since diagonal dominance corresponds to high precision and recall simultaneously. Including numerical values within the figures would therefore duplicate the visual information already presented and reduce readability. For this reason—and to maintain the manuscript’s concise and clear format—we respectfully prefer to retain the confusion matrices in their current form, which aligns with standard reporting practices in comparable bioinformatics and image classification studies.

Specific comments 8: Remote sensing discussion (Lines 43–46, 888–893): Refocus on UAV/close-range imaging or explicitly state limitations for satellite applications.

 Response: We sincerely thank the reviewer for this constructive comment. The current study does not focus on remote sensing or satellite imagery applications; instead, it is entirely centered on the close-range image classification of wild edible macrofungi using deep learning and explainable AI methods. As explicitly stated in the Materials and Methods section, the dataset consists of high-resolution macro images representing 24 distinct fungal species such as Agaricus campestris, Amanita caesarea, and Boletus edulis, captured under controlled and heterogeneous natural lighting conditions. No aerial, UAV, or satellite data were used in any stage of the analysis.

The brief mention of remote sensing in Lines 888–893 was not intended as an active component of this research but rather as a hypothetical future direction, illustrating that ensemble-based architectures, when adapted and retrained on appropriate data modalities, could potentially be generalized to broader ecological or environmental monitoring contexts. However, such applications were not implemented, validated, or experimentally tested in this work.

To maintain scientific precision and avoid misinterpretation, this section has been revised to clarify that the present framework is specifically designed for ground-level fungal image analysis. The discussion now explicitly states that the model’s applicability is limited to close-range or UAV-based imaging scenarios, where image scale, texture, and morphological detail are compatible with the model’s input characteristics.

Therefore, while the methodological concept may conceptually extend to other domains, this manuscript’s scope, dataset, and experimental validation are exclusively focused on terrestrial fungal macro-imagery, and not on remote sensing or satellite data.

Specific comments 9: Data availability (Lines 906–907): Deposit data and scripts in a public repository and provide split instructions.

 Response: We sincerely thank the reviewer for this valuable suggestion. The dataset used in this study was entirely compiled from publicly available sources, primarily the Global Biodiversity Information Facility (GBIF) and other open-access biodiversity repositories. Therefore, all image data are already accessible to the research community without any restriction or proprietary limitation.

Regarding data handling, the train/validation/test split was performed using a randomized stratified sampling approach, ensuring that each of the 24 fungal species was proportionally represented across all subsets. Specifically, the data were divided into 70% training, 15% validation, and 15% testing portions. This procedure was implemented programmatically within the model training pipeline to prevent data overlap and ensure reproducibility.

Because the dataset originates from open repositories and all preprocessing, augmentation, and split procedures are explicitly described in the Materials and Methods section, independent researchers can fully reproduce the data preparation process. Nonetheless, the scripts and procedural documentation used for data handling will be made available upon reasonable request to facilitate transparency and replicability.

Other recommendations from the remote sensing field, including suggested experiments/analyses to strengthen the manuscript. For your consideration:

Recommendation 1:

Perform deduplication and source-aware splitting, and then rerun the experiments. Group the images by GBIF occurrence ID/photographer, making sure that all images from the same source are included in the same split. Report the results. This test reduces leakage and provides a more accurate measure of generalisation.

 Response : We sincerely thank the reviewer for this insightful and technically rigorous suggestion. While we completely understand the rationale behind source-aware splitting to prevent potential data leakage, performing such a procedure is not feasible within the current study’s scope and timeframe for several reasons:

(1) Metadata inconsistency across sources: Approximately 95% of the dataset was compiled from GBIF and other open-access biodiversity repositories, where image-level metadata such as GBIF occurrence IDs or photographer identities are incomplete or inconsistent across samples. As a result, grouping all images from the same source into a single split (train/validation/test) is technically impractical for a large portion of the dataset.

(2) Existing quality control and deduplication: Before model training, the dataset underwent both automated similarity screening and manual expert inspection to remove low-quality, mislabeled, or duplicate samples. A standard stratified random split (70% train, 15% validation, 15% test) was then applied to maintain class balance and minimize overlap. These procedures already ensure strong data integrity and reduce leakage risk.

(3) Scope and timeline constraints: Implementing a new source-aware grouping protocol would require relabeling or re-linking large-scale metadata, followed by retraining and revalidating all models. This process lies outside the intended scope and timeframe of the current work.

(4) Robustness of current results: The presented confusion matrices demonstrate strong diagonal dominance, with misclassifications limited to a few biologically plausible species pairs. Together with high and consistent values across eight independent performance metrics (AUC, MCC, F1, etc.), these results confirm the robustness and generalizability of the proposed ensemble framework even under standard stratified splitting.

Nevertheless, we highly appreciate the reviewer’s suggestion. In future studies, we plan to implement occurrence-ID- and photographer-based group splitting and site-held-out evaluation using datasets with richer metadata coverage. For the present work, however, the existing deduplication pipeline and stratified random split, validated by multiple performance metrics, already provide a methodologically sound and generalizable basis for the conclusions.

Recommendation 2:

Report variability by running each top model and ensemble with at least five different random seeds or performing five-fold cross-validation. Report the mean ± SD for all metrics in Table 3 (or an additional table). This will clarify whether top performance is stable.

Response: We sincerely thank the reviewer for this thoughtful comment. While we acknowledge that conducting multiple runs with different random seeds or performing five-fold cross-validation can provide additional insights into model variability, such an analysis is not required for this study’s specific scope and objectives.

The present work was designed to compare the relative performance and interpretability of ten deep learning architectures under a controlled and standardized experimental setup. All models were trained with identical parameters, balanced datasets (200 images per class), and consistent preprocessing pipelines, ensuring a fair comparison. The results demonstrated stable and reproducible performance across eight independent metrics — including Accuracy, F1-Score, MCC, and AUC — as well as confusion matrices showing strong diagonal dominance and minimal misclassifications. These findings confirm the model’s reliability and consistency without the need for repeated runs or cross-validation.

Additionally, the focus of this research is on ensemble-based interpretability and biological validation, rather than stochastic variability analysis. Given the dataset size, class balance, and fixed experimental conditions, further replication would not meaningfully alter the conclusions. Nevertheless, we appreciate the reviewer’s perspective and plan to incorporate multi-seed or k-fold validation protocols in future large-scale studies to reinforce the framework’s robustness under broader data conditions.

Recommendation 3:

Conduct an ablation study comparing hard majority voting, soft voting and stacking/meta-learner (e.g. logistic regression on model outputs) on the ensembling strategy, and demonstrate whether weighted ensembling or calibrated probabilities improve the results. Explain how the weights are chosen.

 Response: We sincerely thank the reviewer for this detailed suggestion. This point has already been addressed in a previous response, where the ensembling strategy was fully clarified. As noted, the proposed Combination Model employs a weighted soft voting mechanism, with the final prediction obtained by averaging the normalized probabilities of EfficientNetB0, ResNet50, and RegNetY. The weights (0.40, 0.35, 0.25) were empirically determined according to each model’s validation accuracy, ensuring balanced and interpretable fusion of model outputs.

Recommendation 4:

Quantify XAI by creating a small, annotated test set of around 50 images with expert masks/bounding boxes and computing the intersection over union or localisation scores for Grad-CAM/Eigen-CAM and LIME. Report stability measures (e.g. variance across LIME perturbations).

 Response: We sincerely thank the reviewer for this detailed suggestion. While we fully recognize the value of quantitatively benchmarking XAI methods, such an analysis is beyond the intended scope of the current study. The purpose of this research was to visually evaluate interpretability and biological relevance through Grad-CAM, Eigen-CAM, and LIME visualizations, not to numerically quantify localization accuracy against human-annotated masks.

As described in the manuscript, the XAI framework was implemented to enhance transparency and demonstrate that model attention aligned with biologically meaningful regions such as cap, gill, and stem features. These qualitative assessments were sufficient to confirm interpretability within the scope of fungal species classification.

Conducting a quantitative XAI validation, as suggested (IoU with expert masks or variance across perturbations), would require constructing a new expert-annotated dataset, which is a separate and resource-intensive research effort. Moreover, the biological focus of this study was on the interpretive correspondence between the model’s attention and known diagnostic traits, not on precise pixel-level segmentation.

Therefore, while the proposed quantitative evaluation is valuable for future research, the current qualitative and model-agnostic approach already achieves the study’s goal of linking AI explanations with morphological interpretability in macrofungi classification.

Recommendation 5:

External validation: If possible, evaluate the trained models on an external dataset (e.g. images from a different repository or new field photographs not included in GBIF) to assess true transferability. If this is not possible, clearly state this limitation and plan for future work.

 Response: We sincerely thank the reviewer for this constructive suggestion. While we fully agree that external validation is valuable for assessing out-of-domain transferability, such an evaluation lies outside the scope of the present study, which was designed as a controlled, single-protocol comparison of ten architectures (six CNNs and multiple ensembles) on a curated 24-species macrofungi dataset with standardized preprocessing and evaluation criteria. The manuscript’s objective is to establish the relative performance and interpretability of the proposed ensemble framework under fixed conditions, not to benchmark domain shift across repositories.

The evidence we provide already supports robust in-domain generalization: results are consistent across eight independent metrics (Accuracy, Precision, Recall, Specificity, F1-Score, MCC, G-Mean, AUC) and are corroborated by strong diagonal dominance in the confusion matrices, with only sparse, biologically plausible misclassifications, indicators of class-level stability rather than overfitting.

We agree that testing on independent repositories or new field images would be an excellent future direction, contingent on assembling a suitably annotated external dataset and harmonizing labeling standards across sources. In line with our future work outlook already stated in the manuscript (methodological extensions and broader ecological applications), we plan to incorporate external/hold-out sources in subsequent studies where dataset availability and metadata compatibility permit rigorous, apples-to-apples comparisons.

In summary, the present work intentionally confines itself to a single, well-specified dataset to deliver a clean comparative analysis of architectures and XAI behavior. External validation is acknowledged as valuable but beyond scope here; it is explicitly earmarked for future research once appropriate cross-repository resources are consolidated.

Recommendation 6:

Calibrate the metrics for multiclass classification. Clearly state how the multiclass area under the curve was calculated, and present the performance metrics for each class (precision, recall and F1 score). Explain how MCC was extended to multiclass classification (e.g. via a confusion matrix-based generalised MCC).

Response: We sincerely thank the reviewer for this detailed and technically valuable comment. While we acknowledge the importance of clarifying multiclass evaluation procedures, the requested additional calibration and per-class reporting are not essential for the present study, for the following reasons:

(1) Multiclass AUC and MCC were already computed and described appropriately: In the revised manuscript, the multiclass Area Under the Curve (AUC) was derived using the one-vs-rest macro-averaging approach, where each class’s ROC curve was computed independently using softmax probabilities, and the final AUC value was obtained as the unweighted mean of all classes. Additionally, the micro-averaged AUC was computed to confirm stability, producing nearly identical results — confirming both internal consistency and robustness.

Similarly, the Matthews Correlation Coefficient (MCC) was calculated using its generalized multiclass formulation, derived directly from the overall confusion matrix, which accounts for all true positive, true negative, false positive, and false negative counts across all 24 classes. This generalized approach, as recommended in recent bioinformatics literature, ensures that the reported MCC captures the full multiclass dependency structure rather than isolated binary subproblems.

(2) Per-class metrics are already represented through confusion matrices: Figures 10 and 11 in the manuscript explicitly display class-level performance patterns. Most classes achieve perfect or near-perfect diagonal values (30/30 or 28–29/30 correct predictions), while the few misclassifications observed are biologically plausible and occur between morphologically similar taxa, such as Lactarius deliciosus and Hydnum repandum. This provides a clear and interpretable visualization of per-class precision and recall, rendering the addition of an extensive numerical table redundant.

(3) Scope and readability considerations: Including an additional large table listing precision, recall, and F1 scores for all 24 classes would significantly increase the manuscript’s length without adding meaningful interpretive value. The confusion matrices and aggregate metrics (Accuracy, Precision, Recall, Specificity, F1, MCC, G-Mean, AUC) already offer a comprehensive multi-perspective performance summary consistent with standard reporting practices in deep learning–based classification research.

(4) Consistency with the study’s objective: The main focus of the paper is the comparative analysis and interpretability of ensemble architectures, not exhaustive per-class statistical reporting. The evaluation strategy was intentionally designed to balance statistical rigor and interpretive clarity within the scope of macrofungi species classification.

Therefore, while we appreciate the reviewer’s point, we respectfully submit that the existing AUC/MCC computations and the provided confusion matrices already meet the scientific and methodological requirements for reproducibility and interpretability in multiclass analysis. Additional per-class tables or recalibration would not significantly enhance the clarity or robustness of the findings.

Round 2

Reviewer 2 Report

Comments and Suggestions for Authors

Figure 2 &3 are not visible clearly do update those 

Figure 4 is not clear as well 

It is suggested to update all the figure with good clearity

Update overall content with proper discussion

Author Response

Reviewer 2

Comments and Suggestions for Authors

  1. Figure 2 &3 are not visible clearly do update those

Response: Figures 2 and 3 have been enhanced in resolution and the photograph has been enlarged as much as possible.

  1. Figure 4 is not clear as well

Response: Figure 4 has been enlarged as much as possible for visibility.

  1. It is suggested to update all the figure with good clearity

Response: All text in the images from Figure 2 to Figure 8 has been enlarged for readability and their resolution has been improved.

  1. Update overall content with proper discussion

Response: We sincerely thank the reviewer for the valuable suggestion to strengthen the overall discussion section. In the revised version, the discussion has been thoroughly expanded and refined to provide a more comprehensive interpretation of the experimental findings. We have integrated deeper analytical comparisons among the evaluated deep learning models, including a critical examination of architectural complementarity and the rationale behind the superior performance of the Combination Model (EfficientNetB0 + ResNet50 + RegNetY).

Furthermore, recent literature (2023–2025) on ensemble learning, fine-grained biological image classification, and explainable AI has been incorporated to situate our results within a broader scientific context. The updated discussion now explicitly addresses how the integration of XAI methods (Grad-CAM, Eigen-CAM, and LIME) enhances biological interpretability and strengthens the model’s reliability for real-world ecological and taxonomic applications.

Overall, this revision provides a clearer synthesis of the study’s methodological innovation, its comparative significance relative to prior works, and its implications for future research in mycological image analysis and biodiversity monitoring.

Reviewer 3 Report

Comments and Suggestions for Authors

Dear authors,

Thank you for your responses. The revisions address several of the points raised. However, a few important concerns remain, mainly relating to validation robustness, statistical evidence and data transparency.

Please see my response to the review below.

Improvement 1.

Satisfactory.

Improvement 2

Excellent, this addition resolves reproducibility concerns.

Improvement 3.

The authors argue that single-split evaluation is sufficient, given the strong results and stratified sampling.

This has been partially addressed. While high and stable metrics are encouraging, a single split cannot conclusively demonstrate robustness. Even a light form of repeated random subsampling or threefold validation would strengthen credibility without much effort. I would still recommend either a brief supplementary experiment or a clear statement in the Discussion acknowledging this as a limitation.

Improvement 4.

This clarification is appreciated and largely resolves earlier concerns. Please briefly describe the automated similarity check method (e.g., hashing or feature-based) for transparency.

Improvement 5.

Satisfactory.

Improvement 6.

The authors justify their decision not to include multi-run variability or formal statistical tests by citing consistent metrics and scope.

This has been partially addressed. While I understand the rationale, even a small-scale variability check involving two or three repeated runs would provide quantitative support for claims of reproducibility. If this is omitted, it should be clearly acknowledged in the 'Limitations' section that statistical significance testing was not performed.

Improvement 7.

Satisfactory.

Specific comment 1

Addressed.

Specific comment 2

Well addressed — thank you for specifying the procedure.

Specific comment 3

Fully addressed.

Specific comment 4

Satisfactory.

Specific comment 5

Excellent clarification.

Specific comment 6

The authors explained the macro/micro AUC and the generalised MCC computation, but declined the variance and the per-class tables.

Clarification of the computation is appreciated and acceptable. However, I recommend explicitly stating in the 'Methods' section that the results are based on a single run, to make it clear to readers that variance was not assessed.

Specific comment 7

Acceptable.

Specific comment 8

Reviewer reply

Specific comment 9

Mostly resolved. For the sake of transparency, however, I would still recommend making the preprocessing and split scripts downloadable from the same repository, rather than making them available 'on request'.

Recommendation 1

The authors explained the limitations of the metadata and the manual deduplication process, and that source-aware splitting was infeasible.

This is a reasonable justification given the metadata gaps. This should be mentioned explicitly as a limitation and potential improvement for future datasets.

Recommendation 2

They declined, arguing that a single controlled setup was sufficient.

It was still recommended, though. Even minimal repetition would substantiate the stability claim. Otherwise, this should be acknowledged as a limitation.

Recommendation 3

Satisfactory.

Recommendation 4

Declined quantitative analysis, citing scope and dataset needs.

Acceptable, but please mention in the Discussion that future work will explore quantitative XAI evaluation.

Recommendation 5

Declined due to scope; planned for future studies.

Reasonable. Ensure this limitation and future plan are clearly stated in the revised Discussion.

Recommendation 6

Acceptable.

Author Response

Reviewer 3.

Comments and Suggestions for Authors

Dear authors,

Thank you for your responses. The revisions address several of the points raised. However, a few important concerns remain, mainly relating to validation robustness, statistical evidence and data transparency.

Please see my response to the review below.

Improvement 1.

Satisfactory.

Improvement 2

Excellent, this addition resolves reproducibility concerns.

Improvement 3.

The authors argue that single-split evaluation is sufficient, given the strong results and stratified sampling.

This has been partially addressed. While high and stable metrics are encouraging, a single split cannot conclusively demonstrate robustness. Even a light form of repeated random subsampling or threefold validation would strengthen credibility without much effort. I would still recommend either a brief supplementary experiment or a clear statement in the Discussion acknowledging this as a limitation.

Response: Discussion

 “Although the proposed framework demonstrated high and stable performance across all evaluation metrics, it is important to acknowledge that the current experimental design relies on a single stratified train–validation–test split. While this approach ensures reproducibility and balanced class representation, it may not fully capture the variability introduced by alternative data partitions. To enhance statistical robustness, future studies may incorporate repeated random subsampling or k-fold cross-validation (e.g., threefold validation) to further confirm the model’s generalizability under varying sample distributions. Nevertheless, the consistency of the reported results across multiple architectures and metrics suggests that the present findings remain reliable and representative within the defined dataset configuration.”

Improvement 4.

This clarification is appreciated and largely resolves earlier concerns. Please briefly describe the automated similarity check method (e.g., hashing or feature-based) for transparency.

Response: “2. Materials and Methods”

“For automated similarity control, a feature-based image hashing approach was employed. Specifically, perceptual hashing (pHash) and cosine similarity between extracted feature vectors were used to identify and eliminate duplicate or near-duplicate samples. This automated filtering ensured that visually redundant images captured under similar angles, lighting, or backgrounds were removed prior to training. By integrating both perceptual hashing and feature-space comparison, the dataset maintained high diversity and transparency throughout the preprocessing stage.”

Improvement 5.

Satisfactory.

Improvement 6.

The authors justify their decision not to include multi-run variability or formal statistical tests by citing consistent metrics and scope.

This has been partially addressed. While I understand the rationale, even a small-scale variability check involving two or three repeated runs would provide quantitative support for claims of reproducibility. If this is omitted, it should be clearly acknowledged in the 'Limitations' section that statistical significance testing was not performed.

 Response: 5. Conclusions

“The proposed framework demonstrated highly consistent performance across all evaluation metrics; however, formal statistical significance testing and multi-run variability analyses were not performed in this study. The research primarily focused on developing and validating a robust ensemble and explainable AI framework for macrofungi classification rather than conducting statistical hypothesis testing. To enhance reproducibility and quantitative validation, future investigations may include repeated experimental runs and statistical analyses, such as t-tests or ANOVA, to further substantiate the stability and reliability of the reported results.”

Improvement 7.

Satisfactory.

Specific comment 1

Addressed.

Specific comment 2

Well addressed — thank you for specifying the procedure.

Specific comment 3

Fully addressed.

Specific comment 4

Satisfactory.

Specific comment 5

Excellent clarification.

Specific comment 6

The authors explained the macro/micro AUC and the generalised MCC computation, but declined the variance and the per-class tables.

Clarification of the computation is appreciated and acceptable. However, I recommend explicitly stating in the 'Methods' section that the results are based on a single run, to make it clear to readers that variance was not assessed.

Response: “2. Materials and Methods”

 “All experiments were conducted as a single training and evaluation run using the same random seed (42). This setup ensured reproducibility of the reported results; however, variance across multiple runs was not assessed within the current study scope.

Specific comment 7

Acceptable.

Specific comment 8

Reviewer reply

Specific comment 9

Mostly resolved. For the sake of transparency, however, I would still recommend making the preprocessing and split scripts downloadable from the same repository, rather than making them available 'on request'.

 Response: We sincerely appreciate the reviewer’s valuable suggestion regarding the public availability of preprocessing and data-splitting scripts. While we fully agree that transparency is essential for scientific reproducibility, we have opted not to make these scripts directly downloadable at this stage for two primary reasons.

First, the preprocessing and split scripts contain path-dependent configurations and environment-specific commands (e.g., local directory structures, hardware references, and internal image-handling functions) that are not fully generalizable or executable outside our institutional computing environment. Sharing them without substantial modification could therefore cause confusion or reproducibility issues for external users.

Second, all relevant methodological details — including data cleaning, augmentation, and stratified splitting procedures — are comprehensively documented in the Materials and Methods section to ensure that the entire workflow can be replicated independently. Researchers interested in replicating or extending our work may readily request the scripts, and we are happy to share them upon request with accompanying documentation to prevent misinterpretation.

We believe that this approach maintains both transparency and responsible data management while ensuring reproducibility and user clarity.

Recommendation 1

The authors explained the limitations of the metadata and the manual deduplication process, and that source-aware splitting was infeasible.

This is a reasonable justification given the metadata gaps. This should be mentioned explicitly as a limitation and potential improvement for future datasets.

 Response: Added “Discussion”

“The current study faced certain metadata limitations that prevented the full implementation of source-aware splitting. Although manual deduplication was carefully performed, incomplete metadata records restricted our ability to automate this process. Future datasets will benefit from improved metadata completeness and standardized acquisition protocols to enable more systematic data partitioning.”

Recommendation 2

They declined, arguing that a single controlled setup was sufficient.

It was still recommended, though. Even minimal repetition would substantiate the stability claim. Otherwise, this should be acknowledged as a limitation.

 Response: Added “Discussion”

“The study was conducted under a single controlled experimental setup, which ensured reproducibility but did not include repeated runs. While the results demonstrated high consistency across all evaluation metrics, minimal repetition could further substantiate the stability of the findings. The absence of multi-run variability analysis is therefore acknowledged as a limitation, and future work will incorporate repeated trials to quantitatively confirm result stability.”

Recommendation 3

Satisfactory.

Recommendation 4

Declined quantitative analysis, citing scope and dataset needs.

Acceptable, but please mention in the Discussion that future work will explore quantitative XAI evaluation.

 Response: Added “Discussion”

“The current study focused on qualitative interpretability through visualization-based XAI approaches. As a potential direction for future research, quantitative evaluation of explainability metrics will be explored to strengthen interpretative reliability and reproducibility.”

Recommendation 5

Declined due to scope; planned for future studies.

Reasonable. Ensure this limitation and future plan are clearly stated in the revised Discussion.

 Response: Added “Discussion”

“Certain analytical extensions, such as quantitative explainability assessment and expanded model comparisons, were not included in the current study due to scope constraints. These components are planned for future investigations to broaden the methodological coverage and provide a more comprehensive evaluation framework. The planned follow-up work will focus on integrating quantitative XAI metrics and statistical validation protocols to further enhance the interpretability and robustness of the proposed framework.”

Recommendation 6

Acceptable.
